



# Inter-annual, seasonal and diurnal features of the cloud liquid water path over the land surface and various water bodies in Northern Europe as obtained from the satellite observations by the SEVIRI instrument in 2011-2017

Vladimir S. Kostsov[1], Anke Kniffka[2], and Dmitry V. Ionov[1]

[1] Department of Atmospheric Physics, Faculty of Physics, St. Petersburg State University, Russia

[2] Zentrum für Medizin-Meteorologische Forschung, Deutscher Wetterdienst, Freiburg, Germany

*Correspondence to:* Vladimir S. Kostsov (v.kostsov@spbu.ru)

**Abstract.** Liquid water path (LWP) is one of the most important cloud parameters. The knowledge on LWP is critical for
many studies including global and regional climate modelling, weather forecasting, modelling of hydrological cycle and
interactions between different components of the climate system: the atmosphere, the hydrosphere, and the land surface.
Satellite observations by the SEVIRI and AVHRR instruments have already provided the evidences of the systematic
difference between the LWP values derived over the land surface and over the Baltic Sea and major lakes in Northern
Europe during both cold and warm seasons. The goal of the present study is to analyse the phenomenon of the LWP
horizontal inhomogeneities in the vicinity of various water bodies in Northern Europe making focus on the temporal and
spatial variation of LWP. The objects of investigation are water bodies and water areas located in Northern Europe which are
different in size and other characteristics: Gulf of Finland, Gulf of Riga, the Neva River bay, lakes Ladoga, Onega, Peipus,
Pihkva, Ilmen, and Saimaa. The input data are the LWP values of pure liquid-phase clouds derived from the space-borne
observations by the SEVIRI instrument in 2011-2017 during daytime. The study revealed that in general the mean values of
the land-sea LWP gradient are positive during all seasons (larger values over land, smaller values over water surface).
However, the negative gradients were also detected over several relatively small water bodies during cold (winter) season.
The important finding is the positive trend of the land-sea LWP gradient detected within the time period 2011-2017. The
analysis of intra-seasonal features revealed special conditions on the territory of the Gulf of Finland where in June and July
large and moderate positive LWP gradients prevail over negative ones while in August positive and negative gradients are
much smaller (in terms of absolute values) and occur with equal frequency. This result can lead to the conclusion about
possible common physical mechanisms that drive the land-sea LWP difference in the Baltic Sea region at small distances
from the coastline. The diurnal cycle of the LWP land-sea gradient has been detected in June and July while there was no
evidence for it in August. For several specific cases, atmospheric parameters over the mesoscale domain comprising Gulf of
Finland and several lakes have been simulated with the numerical model ICON in limited area and weather prediction mode.





These simulations have clearly demonstrated the LWP land-sea gradient and have pointed out less stability of the atmosphere over land surfaces.

**Keywords:** cloud liquid water path; horizontal inhomogeneity of atmospheric parameters; interactions between the atmosphere and underlying surface, remote sensing; meteorological satellites; SEVIRI

## 1 Introduction

Liquid water path (LWP, the total mass of liquid water droplets in the atmosphere above a unit surface area) is one of the most important cloud parameters. The knowledge on LWP is critical for many studies including global and regional climate modelling, weather forecasting, and modelling of hydrological cycle. The cloud LWP values can be an indicator of the processes of interaction between different components of the climate system: the atmosphere, the hydrosphere, and the land surface. Satellite observations by the SEVIRI and AVHRR instruments have already provided evidence of the systematic

difference between the LWP values derived over the land surface and over the Baltic Sea and major lakes in Northern Europe. The principal findings are the following:

1) During spring and summer the cloud amount over land in this region is larger than the cloud amount over the Baltic Sea and major lakes (Karlsson, 2003).

2) The land-sea LWP gradient is positive during all seasons (larger LWP values over land, smaller LWP values over

water surface) at the estuary of the Neva River in the Gulf of Finland but for the cold season this gradient is noticeably lower than for the warm season (Kostsov et al., 2018). The magnitude of the mean LWP gradient obtained by SEVIRI in this area for the two-year period of 2013-2014 was about 0.040 kg m$^{-2}$ (about 50 % of the mean value over land).

3) In most cases, both SEVIRI and AVHRR instruments demonstrate a similar land-sea LWP gradient, however the

AVHRR data sometimes reveal an inverse (negative) LWP land-sea gradient and unexpected high LWP values over water surface (Kostsov et al., 2019) during the cold season. This phenomenon was attributed by Kostsov et al. (2019) to the artefacts caused by the problems with the ice/snow mask used by the AVHRR retrieval algorithm.

As an illustration of the LWP horizontal inhomogeneities, we present Fig. 1 with the image acquired by the MODIS instrument on May 1, 2013 (North-West part of Russia, parts of Finland and Estonia). The image shows that the Gulf of

Finland and the lakes Ladoga, Onega, Peipus, Pihkva, and Ilmen are cloud-free while there are multiple clouds over the land surface. An interesting feature is the absence of clouds to the East from mentioned water bodies up to a distance of several dozen kilometres. This effect can be explained by the wind flow of the cold near-surface air from water areas eastward. This cold air prevents cloud formation not only above the water areas but also above the land at a certain distance from water bodies. For simplicity, below we use the single term "land-sea gradient" (or "land-sea difference") for designation of the

difference between the LWP value over land and the LWP value over water area regardless of its type (a sea, a gulf, an



estuary, a lake, etc.). For consistency, when talking about satellite measurement pixel over any water area we use the term "sea pixel" (in contrast to "land pixel').

There was an attempt to detect the LWP land-sea differences by means of ground-based microwave observations performed near the coastline of the Gulf of Finland in the vicinity of St.Petersburg, Russia (Kostsov et al., 2020). The
microwave radiometer RPG-HATPRO located 2.5 km from the coastline was remotely probing the air portions over land (zenith viewing geometry) and over water area (off-zenith viewing geometry). The ground-based retrievals of LWP demonstrated that the LWP land-sea gradient existed during all seasons and was positive. This result is in agreement with the space-borne SEVIRI measurements in this region. However, it should be emphasized that the magnitude of the gradient obtained by the ground-based instrument was considerably smaller than detected by SEVIRI. Kostsov et al., (2020) also
reported that the LWP land-sea gradient provided by the ERA-Interim reanalysis for the area and time period under investigation was noticeably smaller than detected by SEVIRI during warm season and, in contrast to the SEVIRI and the ground-based data, it was negative during cold season.

So far, not much attention was paid to the investigation of physical mechanisms which drive the LWP land-sea differences in Northern Europe. The reason for the differences in spring and summer has been suggested by Karlsson (2003):
the inflow of cold water from melting snow and ice is cooling the near-surface atmospheric layer over the water bodies. As a result, in contrast to the land surface, this layer over the water bodies becomes very stable preventing the formation of clouds. This mechanism, however, does not explain the existence of the LWP land-sea gradient during cold season when both land and water surfaces are covered with snow and ice.

In our opinion, the necessary prerequisite for studying physical mechanisms which drive the LWP land-sea
differences in Northern Europe is the special detailed analysis of the LWP data provided by the satellite instruments over various water bodies and over land near these water bodies. The focus should be made on the temporal and spatial variations of LWP at different scales. The goal of the present study is to analyse the phenomenon of the LWP horizontal inhomogeneities in the vicinity of a number of water bodies in Northern Europe which are different in size and in other characteristics: Gulf of Finland, Gulf of Riga, the Neva River bay, Lake Ladoga, Lake Onega, Lake Peipus, Lake Pihkva,
Lake Ilmen, and Lake Saimaa. The study is based on LWP data over Northern Europe obtained from seven years (2011-2017) of the space-borne measurements by the SEVIRI instrument. We try to answer the following main questions:

- What are the statistical distributions of the LWP land-sea gradient during different seasons at different water bodies?
- Does the LWP land-sea gradient always exist during warm and cold season at water bodies with different
properties, and what is its magnitude for large and small water bodies?
- How strong is the inter-annual variability of the LWP land-sea gradient and are there any long-term trends?





- Are there any characteristic features in the diurnal variations of the LWP land-sea gradient (for the day time when space-borne measurements by SEVIRI are available)?
- Is there any correlation between the ice/snow cover period and the magnitude of the LWP gradient?
- Can we distinguish artefacts in the LWP gradient data provided by SEVIRI and, if yes, when and how often do these artefacts appear?

One important remark should be made. The space-borne LWP measurements over land by the SEVIRI instrument were validated extensively by ground-based remote measurements (Roebeling et al., 2008ab; Greuell and Roebeling 2009; Kostsov et al., 2018, 2019). To the best of our knowledge, there were no validations of SEVIRI measurements over water areas and over water bodies covered by ice/snow. The algorithm of the cloud properties' retrieval used by SEVIRI is based on measurements of the reflected solar radiation. During cold season, the LWP retrieval over highly reflective surfaces (snow and ice) is a complicated problem (Musial et al., 2014), and, as a consequence, the retrieval errors can increase. The mechanism of the error amplification is described by Han et al. (1999) and Platnick et al. (2001): (1) multiple reflections occur between a cloud and underlying surface; (2) the increase in reflectance contributed by a cloud is relatively smaller in case of highly reflective underlying surface. The problem becomes more complicated due to the variability of the ice/snow properties. It has been noted by Platnick et al. (2001) that, as shown in a number of studies, the albedo of the sea ice is dependent on several factors, for example on the presence of air bubbles. Besides, if ice is covered with a snow layer greater than several centimetres the overall reflectance is dominated by this snow layer. The melting process can cause the decreases in reflectance. Accounting for the above mentioned reasons, we kept in mind possible effect of the retrieval error amplification while analysing the data obtained during cold season both over land and water bodies.

## 2 Input data

For detailed description of the SEVIRI instrument and the data set containing SEVIRI-derived cloud parameters, we refer to the articles by Stengel et al. (2014) and Benas et al. (2017). The SEVIRI data which we used in the study have the following most important features:

- The geographical region for investigation is centred at St.Petersburg (Russian Federation) and its dimensions are approximately 700 km x 700 km.
- Non-averaged cloud parameters from the level 2 data set covering the time-span 2011 – 2017 were used.
- The ground pixel size in the vicinity of St.Petersburg (at the approx. centre of selected domain) is about 7 km.
- The temporal resolution of the SEVIRI measurements is 15 minutes.
- The data selected for analysis refers to liquid cloud phase only, ice and mixed-phase cloud cases were filtered out.
- The data selected for analysis include all clear sky cases over land and water surfaces which occurred simultaneously and not simultaneously.





- The data acquired when solar zenith angle was larger than 72° were excluded from consideration since the LWP retrieval errors are larger in this case than for smaller solar zenith angles. This is a widely used constraint for data
selection (Roebeling, 2008a; Kim, 2020).

In this study, we consider only the averaged LWP gradient values which are defined as:

$$D = \frac{1}{N}\sum_{k=1}^{N} d_k \,,\qquad(1)$$

where N is the number of pairs of simultaneous measurements of LWP in the sea pixel and in the land pixel during any selected time period, $k$ is the index of this pair of measurements, $d$ is the instantaneous value of the LWP land-sea gradient:

$$d = W_{land} - W_{sea} \,,\qquad(2)$$

where W is the liquid water path value measured in the pixel selected over the land or water area. The quantity d is the small difference of two large quantities. Due to averaging (1), the random component of the gradient error is strongly suppressed and the error of D depends mainly on the biases of LWP over land and over water surfaces. The LWP retrieval errors of the SEVIRI measurements including bias were assessed in several studies. For the complete field of view of SEVIRI (so-called

"the SEVIRI disk"), the bias of the LWP measurements by SEVIRI was specified to be 0.00007 kg/m² for monthly mean values compared to MODIS and the standard deviation amounted to 0.0101 kg m$^{-2}$ (Finkensieper et al. 2016). It is important to note that Finkensieper et al. (2016) reported the overall tendency for SEVIRI to detect higher LWP values over sea and lower LWP values over land, compared to MODIS. However, these differences were not large: the bias of the 45° W-E and S-N area-averaged all-sky (clear cases and cloudy cases together) LWP from SEVIRI with respect to MODIS did not exceed

0.005 kg m$^{-2}$. A comparison with AMSR-E showed a bias of 0.0034 kg m$^{-2}$ over an ocean and a bias-corrected root mean square error of 0.034 kg m$^{-2}$. Roebeling et al. (2008a) compared the LWP data from SEVIRI with the ground-based microwave measurements at three sites for time-series of 4 years and the bias was found to be 0.005 kg/m² in summer and 0.010 kg m$^{-2}$ in winter while the variance was stable with 0.030 kg m$^{-2}$. The comparison of the daily median LWP values obtained by SEVIRI and a ground-based microwave radiometer (Kostsov et al., 2018) has demonstrated the RMS difference

of 0.016 kg m$^{-2}$ for warm season that is considerably lower than the RMS difference for cold season which is 0.048 kg m$^{-2}$. The bias was small and negative for the warm season (-0.003 kg m$^{-2}$), and positive for the cold season (0.002 kg m$^{-2}$). Taking into account all these estimations, we can assume that for warm season, when water areas are not covered by ice and snow, the bias of D is unlikely exceeding 0.010 kg m$^{-2}$. However, the recent study by Kim et al. (2020) should be mentioned also, in which the estimations of the bias of the SEVIRI LWP with respect to LWP from three CloudNet stations (Leipzig,

Lindenberg, and Juelich) were presented for a limited data set and purely liquid phase clouds. These estimations have demonstrated that the bias varies depending on station in a wide range, approximately 0…0.050 kg m$^{-2}$. For cold season, when the surfaces are covered by ice and snow, we do not have any bias estimation, but we can expect the error amplification due to the effect of highly reflective surfaces.





Fig. 2 shows the geographical domain and the water bodies under investigation. As an example, the map of the mean

LWP values for summer period averaged over all 7 years of observations (2011-2017) is also shown: one can see that the LWP land-sea differences exist for all considered water bodies. Despite the fact that Lakes Peipus and Pihkva are located near to each other and are connected, we consider them separately since the area of Lake Pihkva is about three times smaller than the area of Lake Peipus. The present study contains an analysis of the LWP maps of the whole domain and of the LWP gradient values at several specific locations as well. These locations (pairs of the SEVIRI ground pixels) were selected to a

certain extent arbitrarily but we tried to consider the following two cases: the land and the sea pixels of the satellite measurements are far from a coastline with long distance between them (case 1) and they are near a coastline with short distance between them (case 2). For cases 1 and 2, the distance between each of two measurement locations and a coastline is about 40 km and 10 km respectively. So, total distance between measurement locations is about 80 km (case 1) and 20 km (case 2). Also, we tried to select the land and sea pixels in such a way that the line which connects them is oriented close to

the South-North direction. The reason for that was our intention to avoid the influence of the effect of the cold air transport by westerly winds from the water bodies to the land. The westerly and south-westerly winds are predominant in the area of the Gulf of Finland (Monzikova et al., 2013). The geographical coordinates of the selected pixels are presented in Tables 1 and 2 together with other characteristics of the data sets. The data sets are named as ML with one or two numbers. ML stands for "measurement location". The first number identifies the location and the second number identifies long (1) or

short (2) distance, if applicable. Table 2 presents additional measurement locations which have been selected in the course of the analysis to investigate the Gulf of Finland in more detail (see subsequent sections). The locations on the domain map are shown in Fig. 3.

The data selection algorithm worked sequentially for all measurements during 2011-2017 and included checking of the quality flag, solar zenith angle, and the cloud phase. Clear sky cases were included in the data sets as well. For the

175 mentioned specific locations, all data selection criteria should have been satisfied simultaneously for the land pixel measurement and the sea pixel measurement which correspond to a single SEVIRI scan; if not, both the land and sea measurements were filtered out. Finally, we have at our disposal the good quality data on LWP corresponding only to liquid phase clouds and clear sky cases and acquired when solar zenith angle was smaller than 72°. In order to reveal the principal features of the LWP gradient, in this study we limited the analysis by taking into consideration only two time periods. The

180 first period consists of three summer months, and the second period includes February and March. The choice of summer months is evident, but the choice of February and March needs some explanation. Since we explore Northern latitudes, the solar zenith angle is very large during winter resulting in a very small number of SEVIRI measurements suitable for analysis. Such measurements are completely absent in December and January. There are only few measurements in February. Luckily, March is well applicable for analysis due to sufficiently high SZA values and at the same time it can be considered

as a winter month since during March the land is still covered by snow and water bodies are still covered by ice in the region of our interest. Table 1 presents the total number of selected measurements within two mentioned time periods. One can see



that for all measurement locations the number of selected measurements during warm season is large and considerably exceeds the number of measurements during cold season. The datasets for lake Onega contain the smallest amount of data for cold season which is however sufficient for the statistical analysis.

**3 Seasonal and inter-annual features of the LWP gradient**

Figs. 4-6 present statistical distributions of the LWP gradient for specific measurement locations for warm season and for cold season (all years together). It is obvious that if the prevailing positive or negative LWP gradient exists it will show up as the asymmetry of a distribution. One can see such asymmetry in Fig. 4 for measurement locations ML1 and ML2 (the Gulf of Riga and the Gulf of Finland) for both seasons. The central peak which indicates the zero gradients is very sharp in all plots; therefore for better visibility the vertical axes are broken and have different scales in the lower and upper part. Both negative and positive values of the gradient are present in the distributions; however positive values of gradient have the higher frequency of occurrence in all cases. It is interesting that for cold season and small-distance gradients (ML1-2 and ML2-2) the relative frequency of occurrence of negative gradients is negligibly small while it is not the case for large-distance gradients (ML1-1 and ML2-1).

In contrast to the distributions for gulfs of Riga and of Finland (Fig. 4), the LWP gradient statistical distributions for lakes Ladoga and Onega are almost symmetrical except one case: ML3-1 (Lake Ladoga, large-distance gradient) for the warm season (Fig. 5). This result is surprising taking into account the fact that both Lake Ladoga and Lake Onega are featured by low temperature of water in summer, so the land-sea contrast of the near-surface temperature is expected to be noticeable for both of these lakes. Also, attention should be paid to the shape of distributions for the cold season: extremely sharp central peak at the zero gradient and more or less uniform distributions of negative and positive gradients values with very small frequency of occurrence. For the locations ML1 and ML2 (Fig. 4), the distributions for cold season look different, although they are also characterised by a sharp central peak.

Fig. 6 shows the LWP gradient distributions for measurement locations ML5 - ML9. For warm season, we see two distributions with very pronounced asymmetry: for ML5 and for ML9. The case with ML5 (Lake Pihkva) is the most interesting, the negative gradients are almost completely absent for both seasons. For ML9 (the Neva River bay), the negative values of the gradient are present only for warm season. The distributions for the locations M6, M7, and M8 are similar to each other in shape both for warm and cold seasons showing very small asymmetry, however with prevailing positive gradients.

The results presented in Figs. 4-6 can lead to several conclusions. The main conclusion is that the statistical distributions of the LWP gradient can be considerably different for different measurement locations, for different distances between ground pixels, and for different seasons. The most featured distributions are the following:





- Symmetrical with a pronounced peak at zero and Gaussian-shape slopes (warm season);

- Slightly asymmetrical with a pronounced peak at zero and Gaussian-shape slopes (warm season);

- Symmetrical with a sharp peak at zero with low flat slopes (cold season);

– Slightly asymmetrical with a sharp peak at zero with low flat slopes (cold season).

- Strongly asymmetrical and practically without negative values (warm and cold seasons).

To get more insight into the features of the LWP land-sea gradient, we plotted the inter-annual variations of the seasonal-mean LWP gradient for different locations, see Figs. 7 and 8. First of all we pay attention to the standard error of the seasonal-mean values for cold and warm seasons. For better visibility, we did not plot the error bars together with the 225 curves except one case: to give an impression of the magnitude of the error of the mean gradient, in Fig. 8 (c,d) we show the error bars for the results corresponding to ML9. The errors during cold season can reach 0.004 kg m$^{-2}$. The errors for warm season constitute about 0.001…0.002 kg m$^{-2}$.

Analysis of Figs. 7 and 8 leads to the following conclusions:

1) The inter-annual variability of the LWP seasonal-mean land-sea gradient is considerably different for cold season and 230 warm season. The gradient is highly variable for cold season while for warm season in many of considered cases it is close to a constant value.

2) For all considered years, the gradient during warm season is always positive while the situation is different for cold season. Negative values of the gradient are detected for cold seasons of several years at measurement locations ML3, ML4, and ML6, ML7, ML8. We emphasise that the negative gradient was detected at ML2 only once during the cold 235 season of 2013 and its absolute value was negligibly small.

3) If we consider warm season, we can easily classify the gradient values as belonging to three groups: small values (ML3-2, ML4, ML6, ML7, ML8), moderate values (ML1, ML2, ML3-1, ML9), and large values (ML5).

4) The locations ML5 and ML9 are very specific: large positive gradients are detected during both cold and warm seasons.

5) The most important finding is the positive trend of the gradient during 2011-2017 for almost all considered 240 measurement locations. Fig. 9 demonstrates the linear fits for data plotted in Fig. 8. If 2017 is compared to 2011, than for a warm season the increase of the gradient ranges from 0.003 kg m$^{-2}$ to 015 kg m$^{-2}$, and for a cold season the increase ranges from 0.008 kg m$^{-2}$ to 0.012 kg m$^{-2}$. Similar trend of the gradient during 2011-2017 is observed for other locations except ML3-1 (cold season) and ML4-1 and ML3-2 (warm season). In these three cases the trend of the LWP gradient is negative.

To our opinion, the differences in local climate and orography at measurement locations are the most probable reason for the detected differences in the trends, magnitude and inter-annual variations of the seasonal-mean gradients.





## 4 Intra-seasonal features of the LWP gradient

Since the amount of data acquired during the cold season is smaller than during the warm season and almost all data refer to only one month (March) we analysed the intra-seasonal variability of the LWP land-sea gradient only for the warm season. The most robust evidence of the time dependency of the LWP gradient is present at three measurement locations: ML1-2, ML2-2, and ML9, see Fig. 10. This figure shows the daily-mean values of the LWP gradient in June, July and August for all seven years of observations. One can see that in June and July large and moderate positive LWP gradients are prevailing while in August gradients are much smaller (in terms of absolute values) and positive and negative gradients occur with equal frequency. The common feature of two locations ML1-2 and ML-9 is that they are in the vicinity of the estuary of rivers: Western Dvina (Daugava) for ML1-2 and Neva for ML9. The common feature of all three locations is that they correspond to the small-distance gradient and all of them are situated in the gulfs of the Baltic Sea.

The absence of any noticeable time dependence of the LWP gradient during the warm season at other locations is illustrated by Fig. 11. To save space, only four locations are shown which represent moderate gradients (ML3-1), large gradients (ML5) and very small gradients (ML3-2 and ML6). The results for ML7 and ML8 are similar to the results for ML6. The results for ML4-1 and ML4-2 are similar to the results for ML3-2. In order to get more precise estimation of the time dependence rather than demonstrative plots, we present the diagrams of the monthly-mean values of the LWP land-sea gradients for the whole 7-year period in Fig. 12. Different colours in Fig. 12 are used to designate three groups of water bodies: gulfs of the Baltic Sea, large lakes and small lakes. Fig. 12 demonstrates once again that there are three locations which are characterised by very small gradients in August if compared to June and July: ML1-2, ML2-2, and ML9. The locations ML1-1 and ML2-1 may also be attributed to this group; however for these locations the gradient in August is higher than for ML1-2, ML2-2 and ML9. One can also notice that there is another characteristic feature of the inter-seasonal behaviour of the gradient: maximum values are detected in July in most cases (ML1-1, ML1-2, ML2-1, ML3-1, ML4-1, and ML8). No definite time dependence is detected for ML5, ML6, ML7, ML3-2, and ML4-2 (small lakes and small-distance gradients at large lakes).

One can see that all measurement locations with small gradients in August are situated in the Baltic Sea. In order to study this phenomenon in more detail, we selected five more locations in the Gulf of Finland. All these additional locations correspond to a short-distance gradient and are presented in Table 2. Two of them (ML11 and ML12) are close to estuaries of rivers Narva and Luga. ML10 is situated in between ML2-1 and ML11. Two of five additional locations are at the Northern coast of the Gulf of Finland (near Helsinki and Torfyanovka). The intra-seasonal variability of the monthly-mean LWP land-sea gradient averaged over the 7-year period for these additional locations is shown in Fig. 13. One can see that for all additional measurement locations the time dependence of the LWP gradient is the same as for the main locations at the coastline of the Gulf of Finland. One can also see that there are no features which would in one way or another distinguish the gradient obtained near the estuaries of rivers or at the northern coastline with respect to the gradient at other





locations. These results can lead to the conclusion about possible common physical mechanisms that drive the LWP land-sea

difference in the entire Baltic Sea region considered in the present study.

## 5 Diurnal features of LWP

We describe the diurnal variability of the LWP land-sea gradient in terms of mean values which refer to time moments of the SEVIRI measurements (scans are made every 15 min). Averaging was done over seven years of observations. Because of the short period of sun illumination during the cold season, we take only the warm season and study every month separately.

Figs. 14 and 15 demonstrate the diurnal variability obtained for all measurement locations in the Baltic Sea. The error bars indicate the standard error of the mean values. One can see that these bars are relatively large. The reason for this is high variability of the gradient values from one hand and relatively small number of data from the other. We emphasise that for a single time segment the number of available measurements can vary from several dozen to more than one hundred. As a result, the curves in the plots are sometimes oscillating. For better visibility, we do not draw any polynomial approximations

or running average. While analysing the curves we try to mark only the features which are the most pronounced neglecting the oscillations.

First of all we pay attention to the principal feature which has been revealed: the LWP land-sea small-distance gradient is time dependent in June and July and it is time independent and close to zero in August. However, there is one exception for August: the ML13 measurement location (near Helsinki). The general diurnal cycle of the gradient in June and

July is approximately the same for all locations: a smooth increase in the morning until 11…12 h followed by a decrease. There are two features which so far have no explanations. The first one is a very steep decrease of the gradient which occurs just before noon at almost all locations. Another feature is a relatively sharp secondary maximum of the gradient values which is observed at about 16 h at many locations. The most vivid examples of these features are the diurnal cycles in July at ML1-2, ML2-2. As far as the diurnal cycle in August is concerned, we can note that only the ML13 case shows obvious time

dependence of the gradient: the LWP gradient is zero in the morning; it increases until 11 h and than smoothly decreases until the evening when it becomes equal to zero again. It is important to note that for August any conclusions should be made with caution since the gradient values are very small.

The origin of the diurnal cycle of the LWP land-sea gradient is different for June-July and August. In June-July, the diurnal cycle of the gradient is controlled predominantly by the diurnal cycle of the cloud LWP over land since the cloud

LWP over water surface is much smaller from morning till evening. In August, cloud LWPs over land and water surface are comparable and have similar diurnal behaviour, as a result, the LWP gradient is close to zero all the time. The described diurnal cycles of the LWP over land and sea and of the LWP gradient are illustrated in Fig. 16 which refers to measurement locations ML1-2 and ML9.





## 6 Comparison with the reanalysis data

It is interesting and important to compare the obtained statistical characteristics of the LWP land-sea gradient with the data provided by reanalyses. Wright et al. (2020) note that, though cloud fields in reanalyses are essentially model products, many variables, which influence the cloud fields, are altered during the assimilation process. Therefore, detecting differences between the LWP gradient values obtained in experiments and provided by reanalyses can be valuable for identification of possible problems in reanalyses and for future model development. Li et al. (2018) have noted that

*'reanalysis products have become nearly synonymous, in some contexts, with "observations"'*

and they pointed out the necessity

        *'to provide some assessment of this tenuous perception - particularly for quantities such as CLWP and CLWC'*

(here CLWP and CLWC stand for cloud liquid water path and cloud liquid water content).

        In the present study we consider the ERA-Interim and Era5 reanalyses from ECMWF (Dee et al., 2011; Hersbach et

al., 2020). In a previous study (Kostsov et al., 2020) we have noted that the main shortcoming of the comparison of the reanalysis data with the experimental data is the coarse spatial resolution of the reanalysis data: the internal resolution of Era-Interim is 0.75 deg which is about 80 km. For higher resolutions of the Era-Interim data, the interpolation procedure is applied, but the highest recommended resolution is 0.25 deg (28 km). In the present study we have chosen the 28 km resolution of Era-Interim and in this case we could compare the Era-Interim data with the SEVIRI results which correspond

to the large-distance LWP gradient. In contrast to Era-Interim, the Era5 reanalysis has a considerably higher standard resolution which is about 31 km and also it has a higher temporal resolution (1 h versus 6 h in Era-Interim). Besides, there are other improvements in Era5 if compared to Era-Interim. In order to keep consistency with our previous study (Kostsov et al., 2020) and at the same time to account for improvements in reanalysis, we compare the experimental data with both, Era-Interim and Era5. A map showing the geographical locations of the reanalysis grid points used for calculations of the LWP

land-sea gradient is presented in Fig. 17. These locations are designated similar to the corresponding SEVIRI measurement locations: RE1-1, RE2-1, RE3-1, and RE4-1 where RE stands for reanalysis.

        There were three options how to organise all three datasets with respect to temporal sampling. The first and the most evident choice was to select the synchronous data corresponding to 6 h and 12 h UTC from Era-Interim, Era5 and SEVIRI. The second choice was to synchronise the data from reanalyses only (for 6 h and 12 H UTC) and to use the entire SEVIRI

dataset. And the third option was to use the original time sampling for each dataset. We selected the third option due to three main reasons. First of all, our goal was to compare the averaged daytime values for reasonably long time periods, therefore we tried to keep as many initial data as possible. It is especially important for a cold season and SEVIRI data. Second, we did not intend to compare the two versions of reanalysis, but rather wanted to show the agreement or disagreement of reanalyses with the satellite data. In this respect, it was important to keep all improvements which are present in Era5, in

particular its high temporal resolution. And the last but not the least reason: in our previous study (Kostsov et al., 2020) we





used the original temporal resolutions when we compared averaged LWP from three datasets: Era-Interim, SEVIRI and ground-based microwave observations. So, in order to be consistent with our previous results, in the present study the Era-Interim data on LWP corresponding to 6 h and 12 h UTC were collected and averaged over required time periods. For Era5, all data within the time interval 6 h…18 h UTC were collected and averaged over required time periods.

The comparison of the seasonal-mean LWP gradient from SEVIRI with the data from ECMWF reanalyses is presented in Fig. 18 for the period 2011-2017. The most important conclusion, which can be derived from this comparison, is that the magnitude of the LWP gradient provided by the Era5 reanalysis is much larger than provided by Era-Interim. It should be emphasised that this difference is clearly seen for both cold and warm seasons. Moreover, while Era-Interim shows small positive and negative gradients during cold season, the gradients from Era5 during cold season are

predominantly positive, and this finding is rather important. For three locations (ML1-1, ML2-1, and ML3-1), the agreement between SEVIRI and Era5 is much better than between SEVIRI and Era-Interim. For the case with ML2-1, the agreement between the Era5 and SEVIRI data can be even characterised as excellent since the inter-annual behaviour is quantitatively and qualitatively the same. However the attention should be paid to the fact that for all locations Era5 systematically overestimates the LWP gradient from SEVIRI during warm season. In contrast, Era-Interim underestimates the LWP

gradient from SEVIRI during warm season with only one exception at location ML4-1 (Lake Onega): the magnitudes of the SEVIRI and the Era-Interim reanalysis data are in good agreement here.

        Fig. 19 presents the intra-seasonal variability of the LWP land-sea gradient in terms of the monthly-mean (June, July, and August) values derived from the SEVIRI data and from the Era-Interim and Era5 reanalyses. First of all, we pay attention to the fact that Era-Interim reproduces well the effect which was revealed by SEVIRI: the LWP gradient in August

is noticeably smaller than in June and July at a large number of locations (see Section 4). Era-Interim demonstrates this effect even for ML4-1 location which is one of the exceptions. In contrast to that, Era5 shows another type of the intra-seasonal variability: the LWP gradient values in June and August are comparable with each other and at the same time they are smaller that the gradient for July. And as a result, the Era 5 and SEVIRI data demonstrate similar qualitative behaviour for the location ML4-1. However, for all locations and months, Era5 overestimates the LWP gradients if

compared to the SEVIRI data. In contrast to that, Era-Interim strongly underestimates the experimental data, except for location ML4-1.

        Kostsov et al. (2020) have made two conclusions which were based on the cross-comparison of the LWP gradients in the Neva river Bay obtained from SEVIRI with the gradients derived from ground-based microwave observations at the same location, and the gradients provided by Era-Interim near the coastline of the Gulf of Finland at some distance from the

370 Neva Bay:
−    the magnitudes of the LWP gradient provided by ground-based microwave observations and the Era-Interim reanalysis during warm season are in very good quantitative agreement and are considerably smaller than the SEVIRI data;





- the Era-Interim reanalysis data demonstrate a negative LWP gradient during a cold season in contrast to the SEVIRI and ground-based microwave observations.

These conclusions are confirmed by more extensive analysis which was made in the present study. Indeed, Era-Interim underestimates the LWP gradients from SEVIRI for different water bodies at the majority of locations during warm season and provides negative gradients during a cold season. The Era5 reanalysis looks more promising since the Era5 data on LWP gradients are in better agreement with the satellite data despite the fact that they systematically overestimate the experimental data during warm season.

## 7 LWP gradient simulations with the ICON model

The ICON modelling framework (ICOsahedral Nonhydrostatic weather and climate model) is a unified next-generation global numerical weather prediction and climate modelling system which has been introduced as operational forecast system of the German Weather Service in January 2015 (ICON, 2021). ICON has proven to be a powerful tool for studying cloud formation processes, for example Costa-Surós et al. (2020) used the special configuration ICON-LEM (Large Eddy Model)

to study the response of clouds to realistic anthropogenic perturbations in aerosols. Costa-Surós et al. (2020) noted the high computational cost of running ICON in the highest resolution mode and limited their research by a simulation of only one single day. Nevertheless, their work showed the great potential of combining high resolution simulations with a large set of observations for studying cloud-aerosol interactions. In the present study we selected ICON as an optimal tool for simulations of the LWP land-sea differences due to two main reasons. First, its spatial resolution is sufficient for taking into

account water bodies of a relatively small size, and, second, ICON incorporates special parameterization scheme for lakes "Flake". As it is stated in the ICON tutorial (ICON Tutorial, 2021), in contrast to oceans and seas, the water surface temperature for lakes should not be kept constant over the entire forecast period. Diurnal variations of the water surface temperature can reach several degrees and for frozen lakes the diurnal variations of the ice surface temperature may exceed ten degrees. The details of "Flake" can be found in ICON Tutorial (2021), references therein, and at http://lakemodel.net

(last access 2 April, 2021). It is important to note that generally no observational data are assimilated into FLake, i.e., the evolution of the lake temperature, the lake freeze-up, and break-up of ice occur freely during the ICON runs except for Laurentian Great Lakes of North America (ICON Tutorial, 2021).

Because of the high computational cost, we limited our simulations to only a few ICON runs. We simulated two single days 25 July 2015 and 12 August 2016 starting from 00 UTC to 00 UTC on the following day but for the relatively

large domain comprising the Gulf of Finland and several lakes. Our primary goal was to evaluate how ICON reproduces the LWP land-sea difference and to analyse the characteristic features of atmospheric parameters over water and land surfaces. Besides, we tried to find an explanation for the intra-seasonal variation of the LWP gradient, namely for the high LWP gradient in July and the low LWP gradient in August over the territory of Gulf of Finland as observed by SEVIRI (see Section 4 above). Selection of days for modelling was done on the basis of analysis of the cloud images of the Neva Bay and





part of Gulf of Finland acquired by the MODIS instrument. The time of MODIS overpass over this region is about 11:45 UTC (13:45 local solar time). The MODIS image for 25 July 2015 is presented in Fig. 20a and clearly demonstrates the presence of clouds over land and the absence of clouds over water. Such a situation is typical for June and July as shown by SEVIRI. The MODIS image for 12 August 2016 is presented in Fig. 21a and demonstrates clouds, which are uniformly distributed over land and water surfaces which is typical for August according to the results of the SEVIRI observations.

Fig. 20b demonstrates the LWP maps obtained from the simulations by ICON for 25 July 2015. The time span of the model results is 11…16 h UTC that means 13…18 h local solar time. The model maps vividly demonstrate the evolution of cloudiness over land in the afternoon. Clouds were absent in the morning both over land and water surfaces. At about 11 UTC the cloud formation process started over land around the Gulf of Finland. Maximal cloudiness existed during the time period 13…14 h UTC, and afterwards the clouds began to disappear. During the whole day there were no clouds over water
surfaces. So, simulations for this single day are in good agreement with the diurnal cycle for July and June derived from the SEVIRI observations: the evolution of clouds over land and the absence of clouds over water bodies. Fig. 21b demonstrates the LWP maps obtained from the simulations by ICON for 12 August 2016. In contrast to the maps for 25 July 2015, there is only a weak diurnal cycle within the time period 11…16 h UTC. Clouds are distributed mostly in the area at the southern coastline of Gulf of Finland over land and over water as well and no significant change in terms of cloud cover occurs during
the considered time period.

The difference in the cloud cover for the two modelled days can be explained if we look at the near-surface (at 2 m) temperature maps which are presented in Fig. 22 for 25 July 2015 and in Fig. 23 for 12 August 2016. For 25 July 2015, we see a strong land-sea contrast of the surface temperature. During the entire time period from 11 to 16 h the surface temperature over Gulf of Finland was 15 to 17 °C while it was higher than 20 °C over land. But there is a much smaller
land-sea contrast in surface temperature on 12 August 2016: the temperature is in the range 14…17 °C over land and water. Moreover, at 15 h and 16 h the near-surface temperature over water at the southern coastline of Gulf of Finland was even higher than over land. These observations indicate that the land-sea contrast of the near-surface temperature is the major driver of the cloud cover land-sea differences in cases where strong frontal systems are absent and, hence, of the LWP land-sea differences. This conclusion is in agreement with the statement which has been made by Karlsson (2003) and refers to
the stability of the atmosphere caused by cooling of the low atmospheric layer by cold water of the Baltic Sea and major lakes in spring and early summer. This mechanism seems to be of general character and works during other seasons as well.

In order to shed light on specific features of the atmospheric state over water and land surfaces, we have computed the averaged vertical profiles of several parameters (cloud cover, specific cloud water content, relative humidity, and turbulent kinetic energy) for two sub-domains within the general domain for modelling. One of these sub-domains comprises water
surface only and another sub-domain comprises the adjacent land territory. The averaged profiles with a brief description of the involved processes are presented in Appendix A.





To our opinion, the results of the ICON simulations confirm our initial assumptions. For 25 July 2015, they vividly demonstrate the LWP land-sea gradient and point out less stability of the atmosphere over land surfaces. Using ICON, we simulated the day in August (12 August 2016) trying to detect the difference in cloud formation in July and in August which

has been revealed by SEVIRI (large gradient in July and small one in August due to appearance of clouds over water). And this effect has been indeed reproduced in our simulations and is explained in more detail in Appendix A. Of course, one should keep in mind that the days for modelling have been specially selected as "typical" days on the basis of analysis of the MODIS images.

**Summary and conclusion**

Studying interactions between different components of the climate system (the atmosphere, the hydrosphere, and the land surface) is a very interesting scientific task due to involvement of many processes of different kind in these interactions. Cloud amount and cloud liquid water path (LWP) are quantities which can be the indicators of some of these processes relevant, in particular, to an exchange of moisture and heat between a surface and the atmosphere. The goal of the present study is to analyse the phenomenon of the LWP horizontal inhomogeneities in the vicinity of various water bodies in

Northern Europe making focus on the temporal and spatial variation of LWP. The motivation for the study is the desire to explore in more detail the systematic difference between LWP values over the land surface and over the Baltic Sea and major lakes in Northern Europe which was revealed previously from the satellite observations by the SEVIRI and AVHRR instruments.

The input data are the LWP values of pure liquid-phase clouds derived from the space-borne observations by the

SEVIRI instrument in 2011-2017 during daytime in the northern part of Europe. The geographical region for investigation is centred at St.Petersburg (Russian Federation) and its dimensions are approximately 700 km x 700 km. The SEVIRI ground pixel size at the approx. centre of selected domain is about 7 km. The temporal resolution of the SEVIRI measurements is 15 minutes. The data selected for analysis refers to liquid cloud phase only, ice and mixed-phase cloud cases were filtered out. The data selected for the analysis include all clear sky cases over land and water surfaces which occurred simultaneously and

not simultaneously as well. The objects of investigation are water bodies which are different in size and other characteristics: Gulf of Finland, Gulf of Riga, the Neva River bay, lakes Ladoga, Onega, Peipus, Pihkva, Ilmen, and Saimaa. A set of measurement locations has been selected for these water bodies and two cases have been considered: the land and the sea pixels of the satellite measurements are far from a coastline with long distance between them (case 1) and they are near a coastline with small distance between them (case 2). For cases 1 and 2, the distance between each of two measurement

locations and a coastline is about 40 km and 10 km respectively.

The statistical distributions and the mean values of the LWP gradient averaged over large number of measurements have been used for analysis. The analysis has shown the following:





1)   The statistical distributions of the LWP gradient vary from symmetrical ones with the pronounced peak at zero value and Gaussian-shape slopes to strongly asymmetrical ones without negative values.

2)   The inter-annual variability of the LWP seasonal-mean land-sea gradient is considerably different for cold season and warm season. The gradient is highly variable for cold season while for warm season in many of considered cases it is close to a constant value. For all analysed years, the gradient during warm season is always positive while the situation is different for cold season.

3)   The most important finding is the positive trend of the LWP gradient during 2011-2017 for almost all considered measurement locations. To our opinion, the differences in local climate and orography at measurement locations are the most probable reason for the detected differences in the trends, magnitude and inter-annual variations of the seasonal-mean gradients.

4)   The intra-seasonal variability of the LWP land-sea gradient has been analysed only for a warm season because of lack of data for a cold season. It has been found that over Baltic Sea in June and July large and moderate positive LWP gradients are prevailing while in August positive and negative gradients are much smaller (in terms of absolute values) and occur with equal frequency.

5)   For analysis of diurnal features of the LWP gradient, we considered only the warm season because of the short period of sun illumination during the cold season. The principal feature which has been revealed is the following: the LWP land-sea small-distance gradient is time dependent in June and July and it is time independent and close to zero in August. The general diurnal cycle of the gradient in June and July is approximately the same for all locations: a smooth increase in the morning until 11…12 h local solar time followed by a decrease. The origin of the diurnal cycle of the LWP land-sea gradient is different for June-July and August. In June-July, the diurnal cycle of the gradient is controlled predominantly by the diurnal cycle of the cloud LWP over land since the cloud LWP over water surface is much smaller from morning till evening. In August, cloud LWPs over land and water surface are comparable and have similar diurnal behaviour, as a result, the LWP gradient is close to zero all the time.

6)   There are two features of the diurnal cycle which so far have no explanations. The first one is a very steep decrease of the LWP gradient values which occurs just before noon at almost all locations. Another feature is a relatively sharp secondary maximum of the gradient values which is observed at about 16 h at many locations. The most vivid examples of these features are the diurnal cycles in July at Gulf of Riga and Gulf of Finland. Probably, these two features can be instrumental artefacts caused by some peculiarities of observations and retrieval procedure.

7)   The obtained statistical characteristics of the LWP land-sea gradient have been compared with the data provided by reanalyses ERA-Interim and Era5 from ECMWF. Era-Interim underestimates the LWP gradients from SEVIRI for different water bodies at the majority of locations during warm season and provides negative gradients during a cold season. The Era5 reanalysis looks more promising since the Era5 data on LWP gradients are in better qualitative and quantitative agreement with the satellite data despite the fact that they systematically overestimate the experimental data during warm season.





8)  The ICON model simulations of atmospheric parameters over the mesoscale domain comprising Gulf of Finland and several lakes have been done for two specific cases/days: without (1) and with (2) clouds over water area. These cases have been selected on the basis of the analysis of the MODIS cloud images. The simulations for case 1 have clearly demonstrated the LWP land-sea gradient and have pointed out less stability of the atmosphere over land surfaces. Simulations for this day are in good agreement with the diurnal cycle for July and June derived from the SEVIRI observations: the evolution of clouds over land and the absence of clouds over water bodies. Simulations for case 2 have been also successful and have demonstrated cloudiness over land and over water. The ICON simulations point out the land-sea contrast of the near-surface temperature as the major driver of the cloud cover land-sea differences causing different boundary layer stratifications and, hence, of the LWP land-sea differences.

There are still many questions to which we have not yet found answers. In particular, we did not investigate a correlation between the ice/snow cover period and the magnitude of the LWP gradient since the amount of the SEVIRI data in winter is extremely small (the SEVIRI observations are possible only under sun illumination conditions). The fact that the maximal LWP gradients are observed in the vicinity of the Neva River bay and of the Lake Peipus also needs an explanation. And finally, our study has demonstrated large variability of the LWP gradient features for different locations, especially for lakes, and this variability, as we believe, also requires further analysis. Nevertheless, to our opinion, the most important findings of the present work are the positive long-term (7-year) trend of the magnitude of the LWP land-sea gradient and the so-called "August anomaly": the absence of the LWP gradient in August in contrast to June and July. It should be emphasised that this "August anomaly" is strictly limited to Gulf of Finland.

**Data availability**

The LWP data derived from the SEVIRI observations are available at https://www.cmsaf.eu, last access: 9 April 2021 (EUMETSAT CM SAF, 2021).

**Author contributions**

VSK conceived the study and was in charge of preparing the draft of the manuscript. DVI and AK were responsible for data selection and analysis. AK made all simulations with the ICON model. VSK, DVI and AK together interpreted the results, made the detailed analysis, reviewed and edited the manuscript.

**Competing interests**

The authors declare that they have no conflict of interest.



**Funding**

This research has been supported by Russian Foundation for Basic Research through the project No. 19-05-00372.

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





**Table 1: Geographical coordinates the total number of LWP measurements selected for analysis, and the data set designation.**

| Water body | Data set | Coordinates of measurement locations | | Total number of measurements | | Distance between pixels |
|---|---|---|---|---|---|---|
| | | sea | land | warm season (June-August) | cold season (February-March) | |
| 1. Gulf of Riga | ML1-1 | 57.366416N, 23.664965E | 56.765244N, 22.910569E | 15924 | 4655 | LD** |
| | ML1-2 | 57.139038N, 23.385504E | 56.993382N, 23.193808E | 17884 | 5065 | SD*** |
| 2. Gulf of Finland | ML2-1 | 59.857132 N, 25.299348 E | 59.144346 N, 25.536151 E | 15478 | 3824 | LD |
| | ML2-2 | 59.583547 N, 25.387536 E | 59.404679 N, 25.452862 E | 17139 | 4196 | SD |
| 3. Lake Ladoga | ML3-1 | 60.487217 N, 32.255603 E | 59.787115 N, 32.622665 E | 16277 | 3596 | LD |
| | ML3-2 | 60.228835 N, 32.387668 E | 60.056733 N, 32.484774 E | 17169 | 3704 | SD |
| 4. Lake Onega | ML4-1 | 61.389299 N, 35.810093 E | 60.726050 N, 36.389155 E | 15034 | 2934 | LD |
| | ML4-2 | 61.147303 N, 36.018435 E | 60.978995 N, 36.167049 E | 15874 | 2679 | SD |
| 5. Lake Peipus | ML5 | 58.906952 N, 27.367954 E | 59.091926 N, 27.350632 E | 16457 | 4040 | SD |
| 6. Lake Pihkva | ML6 | 57.940914 N, 27.992420 E | 57.763346 N, 27.942981 E | 17305 | 4532 | SD |
| 7. Lake Ilmen | ML7 | 58.235672 N, 31.173657 E | 58.058871 N, 31.117336 E | 18006 | 4454 | SD |
| 8. Lake Saimaa | ML8 | 61.222858 N, 28.439451 E | 61.042522 N, 28.468290 E | 16421 | 3662 | SD |
| 9. The Neva River bay* | ML9 | 59.957250 N, 29.913798 E | 59.880953 N, 29.826097 E | 18001 | 4282 | SD |

\* Corresponds to ground-based microwave measurements from the study by Kostsov et al.(2020)
\*\* large distance (about 80 km, see text)
\*\*\* small distance (about 20 km, see text)





**Table 2: Same as Table 1 but for additional measurement locations.**

| Water body | Data set | Coordinates of measurement locations | | Total number of measurements | | Distance between pixels |
|---|---|---|---|---|---|---|
| | | sea | land | warm season (June-August) | cold season (February-March) | |
| 10. Gulf of Finland, Kalvi | ML10 | 59.580558 N, 26.846173 E | 59.407040 N, 26.723764 E | 17279 | 4370 | SD |
| 11. The Narva River bay | ML11 | 59.490327 N, 27.810318 E | 59.308683 N, 27.811472 E | 16931 | 4460 | SD |
| 12. The Luga river bay | ML12 | 59.766497 N, 28.324202 E | 59.586924 N, 28.325356 E | 17588 | 4312 | SD |
| 13. Gulf of Finland, Helsinki | ML13 | 60.167451 N, 25.310649 E | 60.341055 N, 25.226349 E | 17146 | 4059 | SD |
| 14. Gulf of Finland, Torfyanovka | ML14 | 60.428589 N, 27.962023 E | 60.609377 N, 27.918291 E | 16996 | 3938 | SD |



**Table 3: Summary for analysis of the diurnal variation of the LWP land-sea gradient.**

| Measurement location ID | Description | The evidence for diurnal variation of the LWP land-sea gradient: detected (+) / not detected (-) | | |
| --- | --- | --- | --- | --- |
| | | June | July | August |
| ML1-1 | Baltic Sea | + | + | - |
| ML1-2 | | + | + | - |
| ML2-1 | | + | + | + |
| ML2-2 | | + | + | - |
| ML9 | | + | + | - |
| ML10 | | + | + | - |
| ML11 | | + | + | - |
| ML12 | | + | + | - |
| ML13 | | + | + | + |
| ML14 | | + | + | - |
| ML3-1 | Big lakes | + | + | + |
| ML3-2 | | - | - | - |
| ML4-1 | | + | + | - |
| ML4-2 | | - | - | - |
| ML5 | Small lakes | + | + | + |
| ML6 | | - | - | - |
| ML7 | | - | - | - |
| ML8 | | - | - | - |



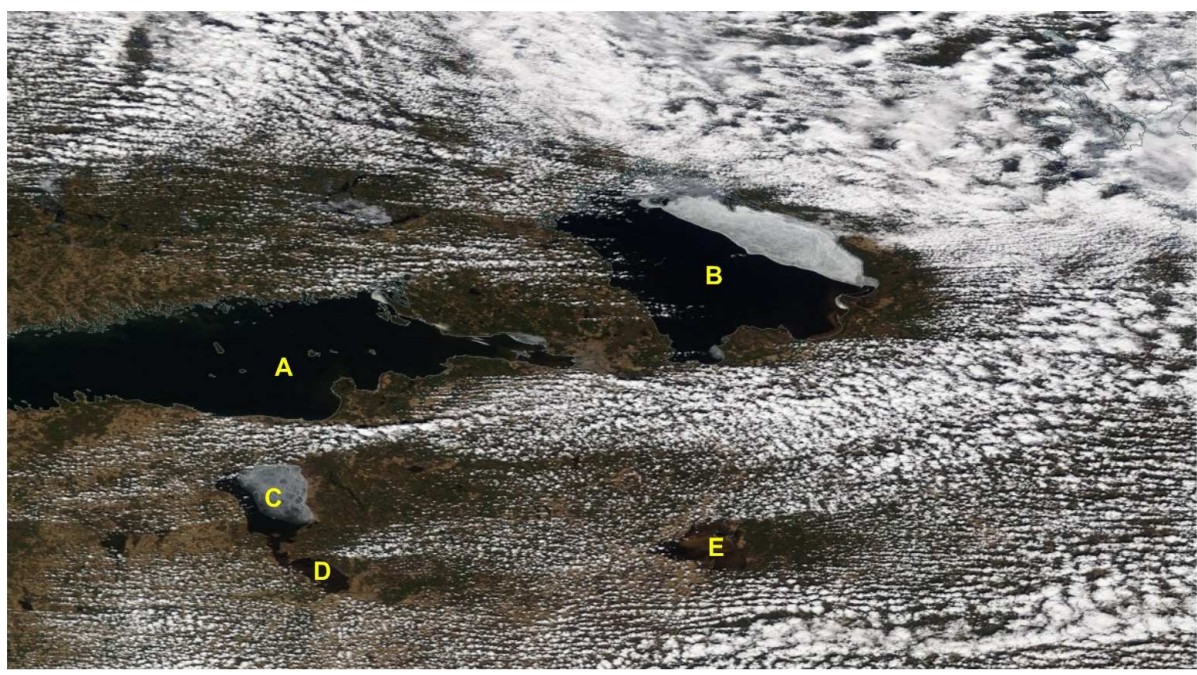

**Figure 1: The MODIS image acquired on May 1, 2013 which shows Gulf of Finland (A), Lake Ladoga (B), Lake Peipus (C), Lake Pihkva (D), Lake Ilmen (E).**

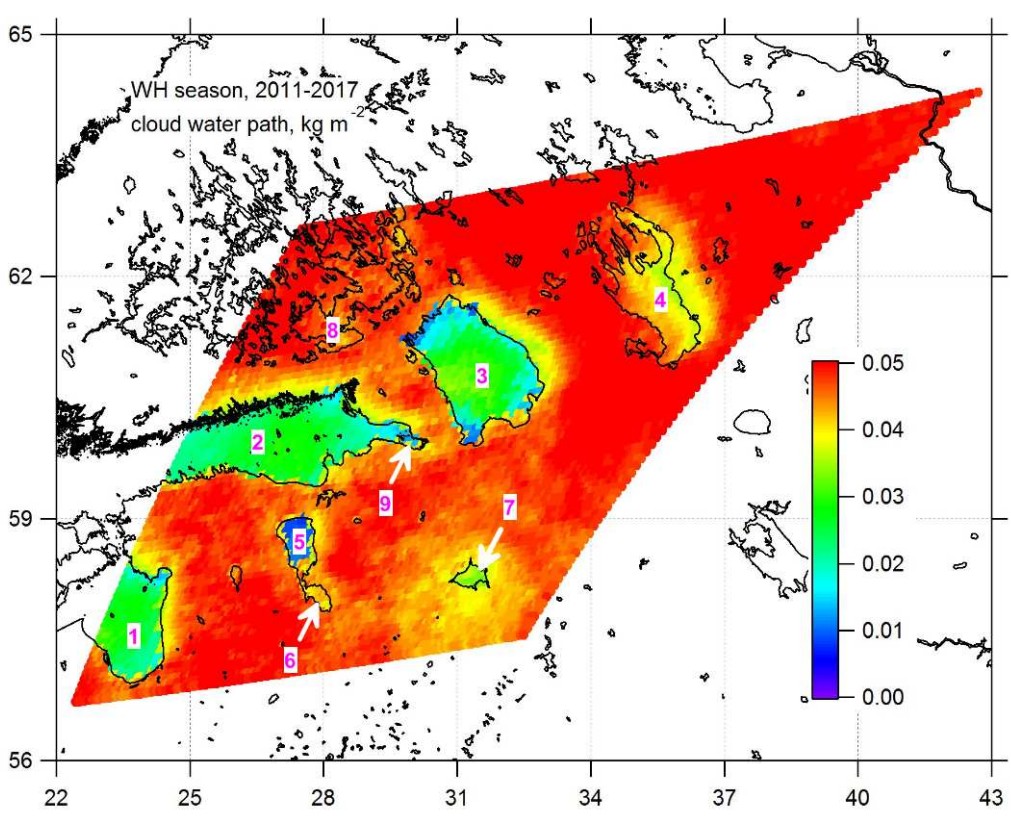

**Figure 2:** **The geographical domain and the water bodies under investigation: Gulf of Riga (1), Gulf of Finland (2), Lake Ladoga (3), Lake Onega (4), Lake Peipus (5), Lake Pihkva (6), Lake Ilmen (7), Lake Saimaa (8) , and the Neva River bay (9). This example map demonstrates the SEVIRI-derived mean LWP for summer 2015. Vector shoreline data: (GSHHG, 2017).**



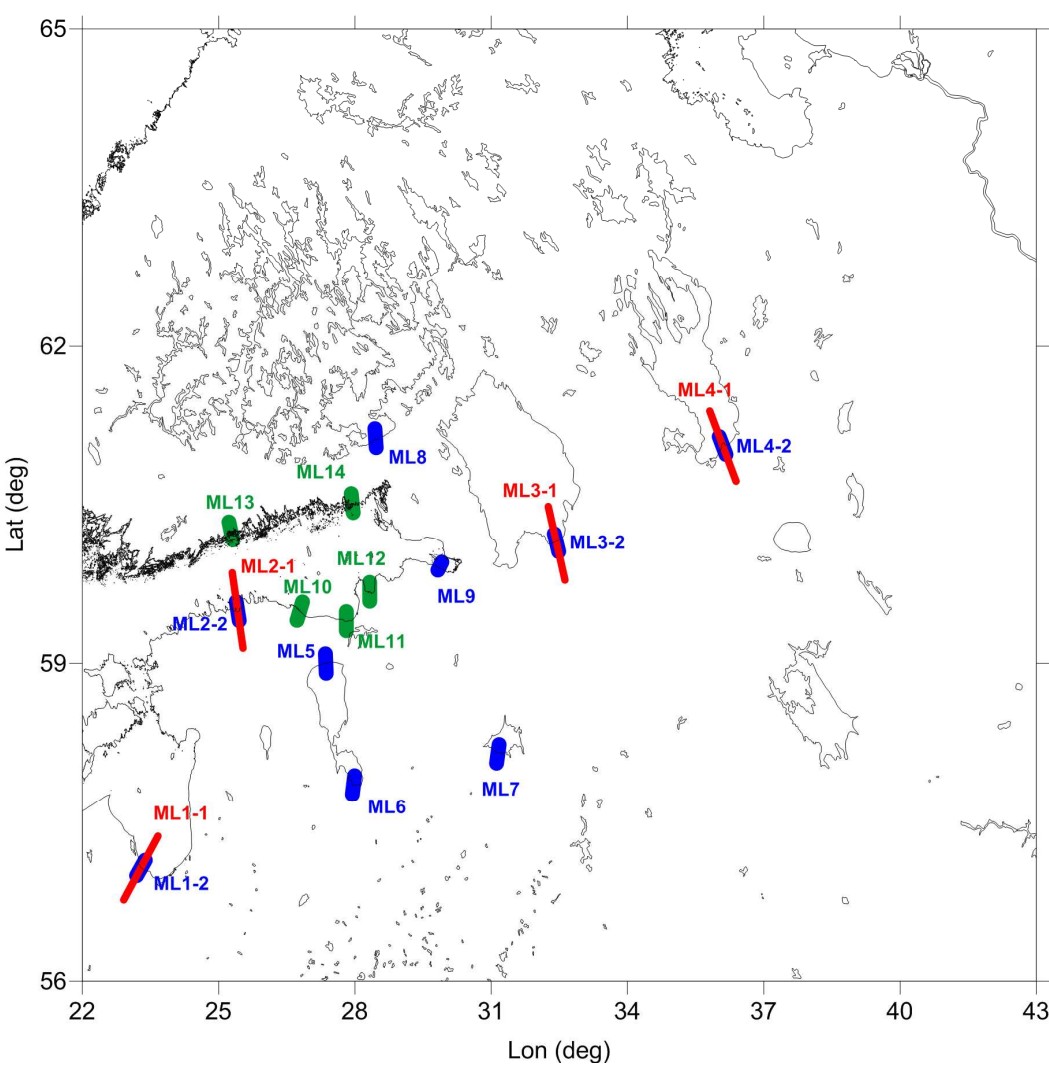

**Figure 3:** **The geographical domain and the specific locations of the LWP gradient measurements (shown as lines which connect ground pixels of the SEVIRI measurements, red – large distance gradient, blue and green – small distance gradient). Green colour is used to mark additional measurement locations in the Gulf of Finland, see text. The colour letters with numbers indicate the corresponding datasets (see Tables 1 and 2). Vector shoreline data: (GSHHG, 2017).**



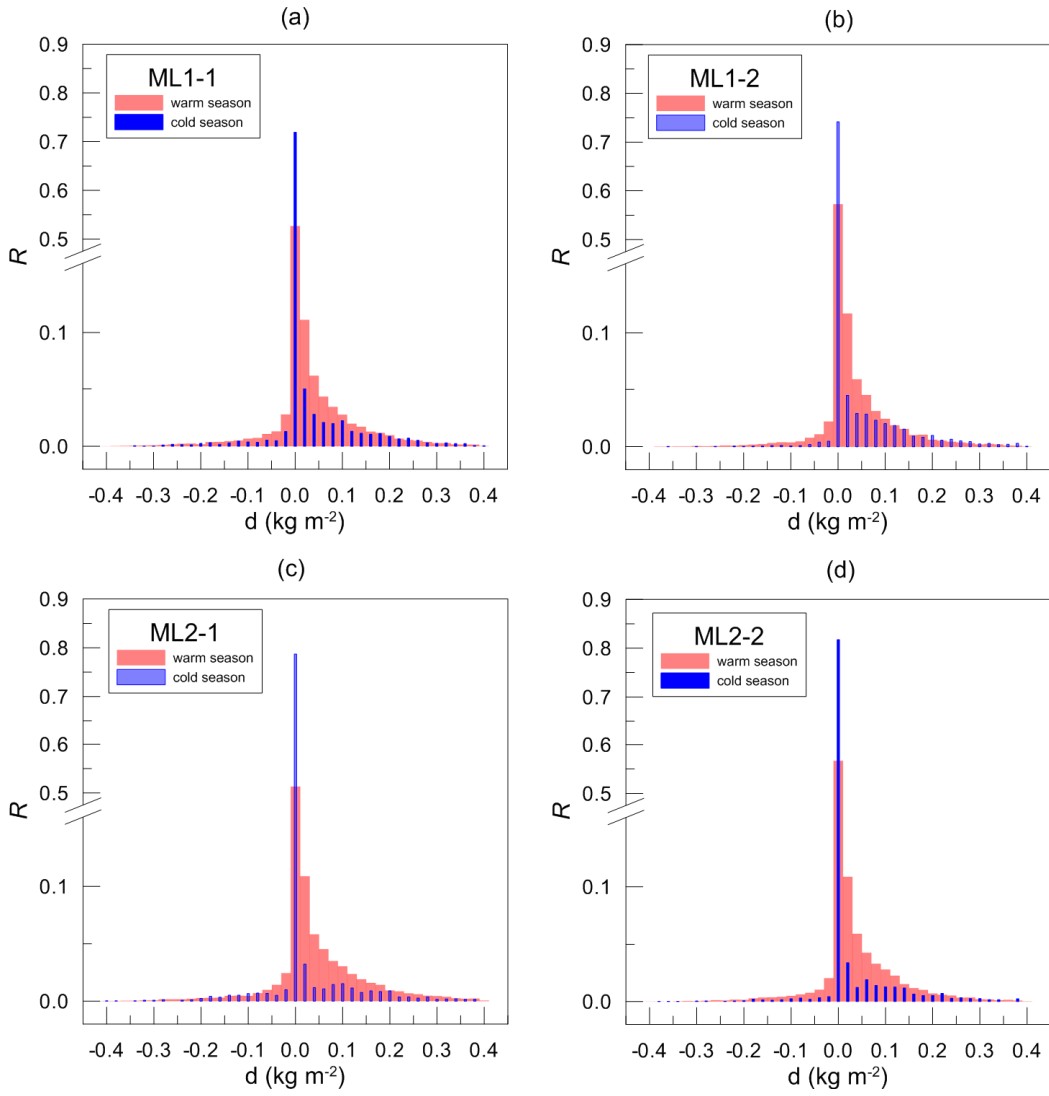

**Figure 4: Statistical distributions (in terms of relative frequency of occurrence R) of the LWP gradient values for measurement locations ML1 and ML2 and different seasons. Please note that for better visibility the vertical axes are broken and have different scaling in the lower and upper part.**

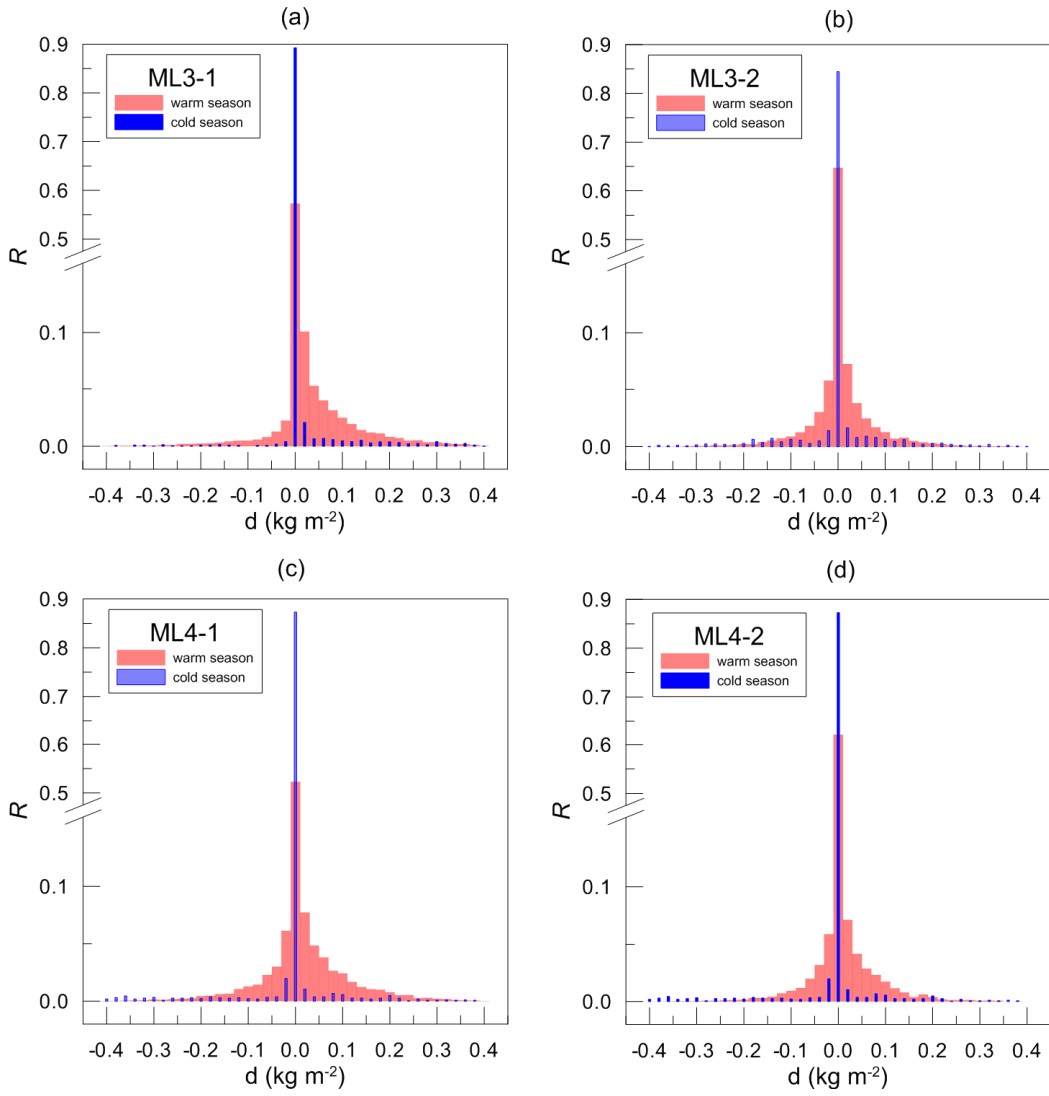

**Figure 5: Statistical distributions (in terms of relative frequency of occurrence R) of the LWP gradient values for measurement locations ML3 and ML4 and different seasons. Please note that for better visibility the vertical axes are broken and have different scaling in the lower and upper part.**



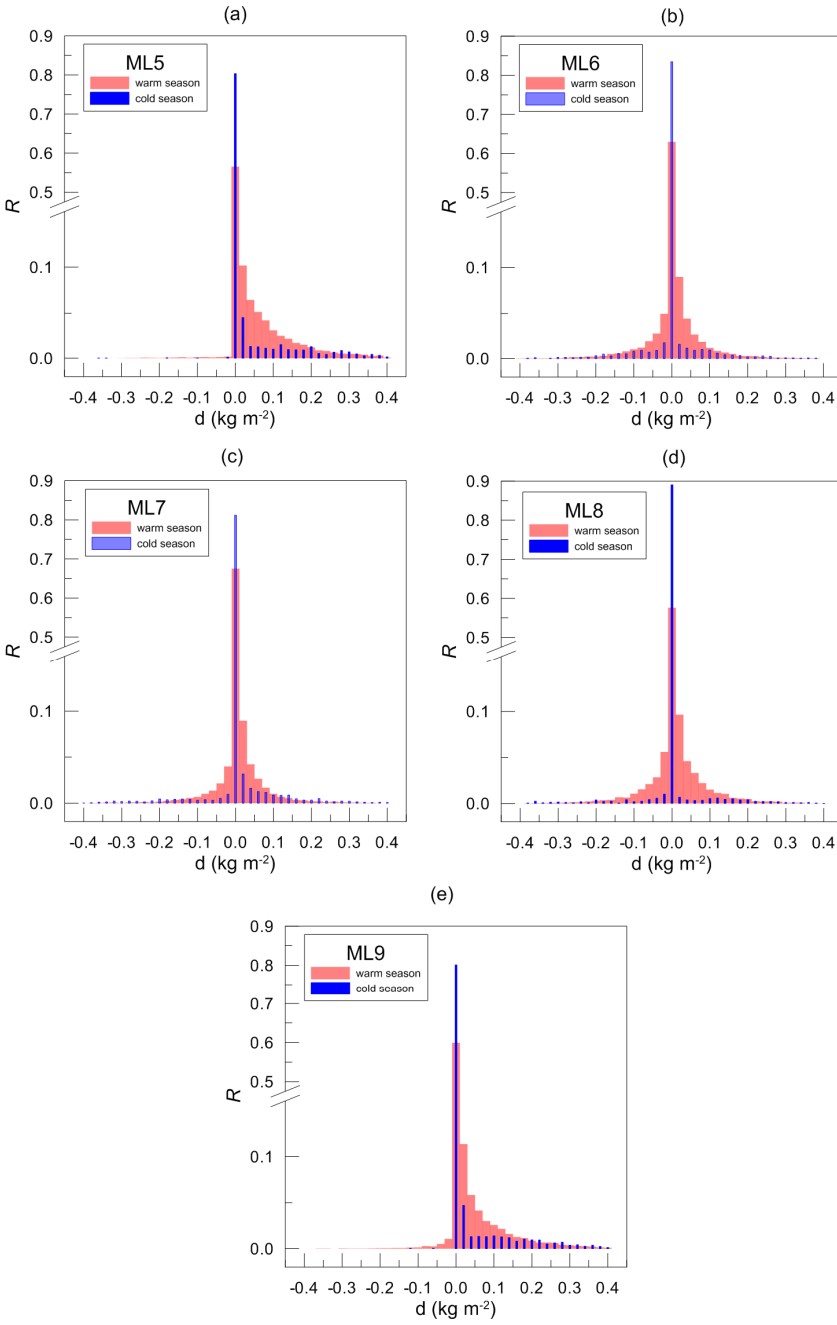

**Figure 6: Statistical distributions (in terms of relative frequency of occurrence R) of the LWP gradient values for measurement locations ML5 – ML9 and different seasons. Please note that for better visibility the vertical axes are broken and have different scaling in the lower and upper part.**





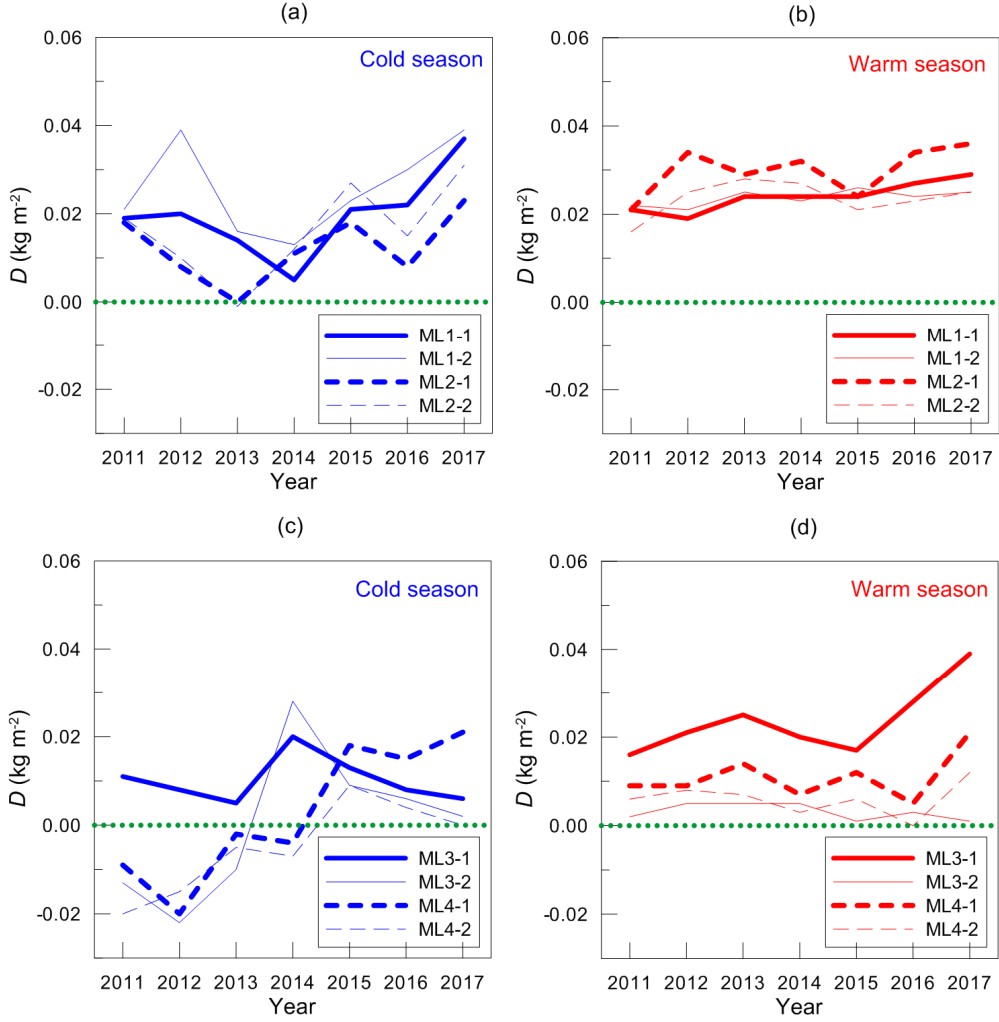

**Figure 7: Inter-annual variability of the seasonal-mean LWP land-sea gradient for measurement locations ML1 – ML4 and different seasons. Green dots indicate the zero-gradient line.**

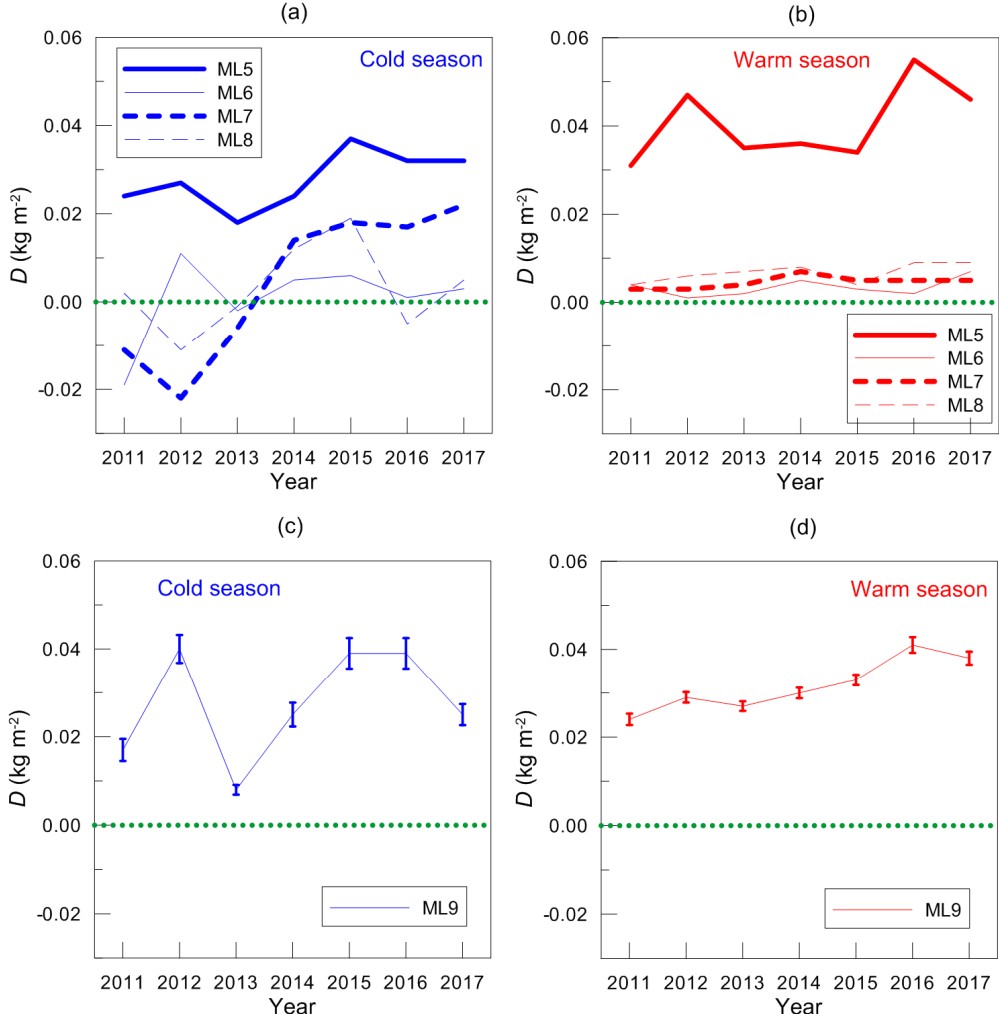

**Figure 8: Inter-annual variations of the seasonal-mean LWP land-sea gradient for measurement locations ML5 – ML9 and different seasons. Green dots indicate the zero-gradient line. Panels (c) and (d) demonstrate not only the seasonal-mean values but also their standard errors.**





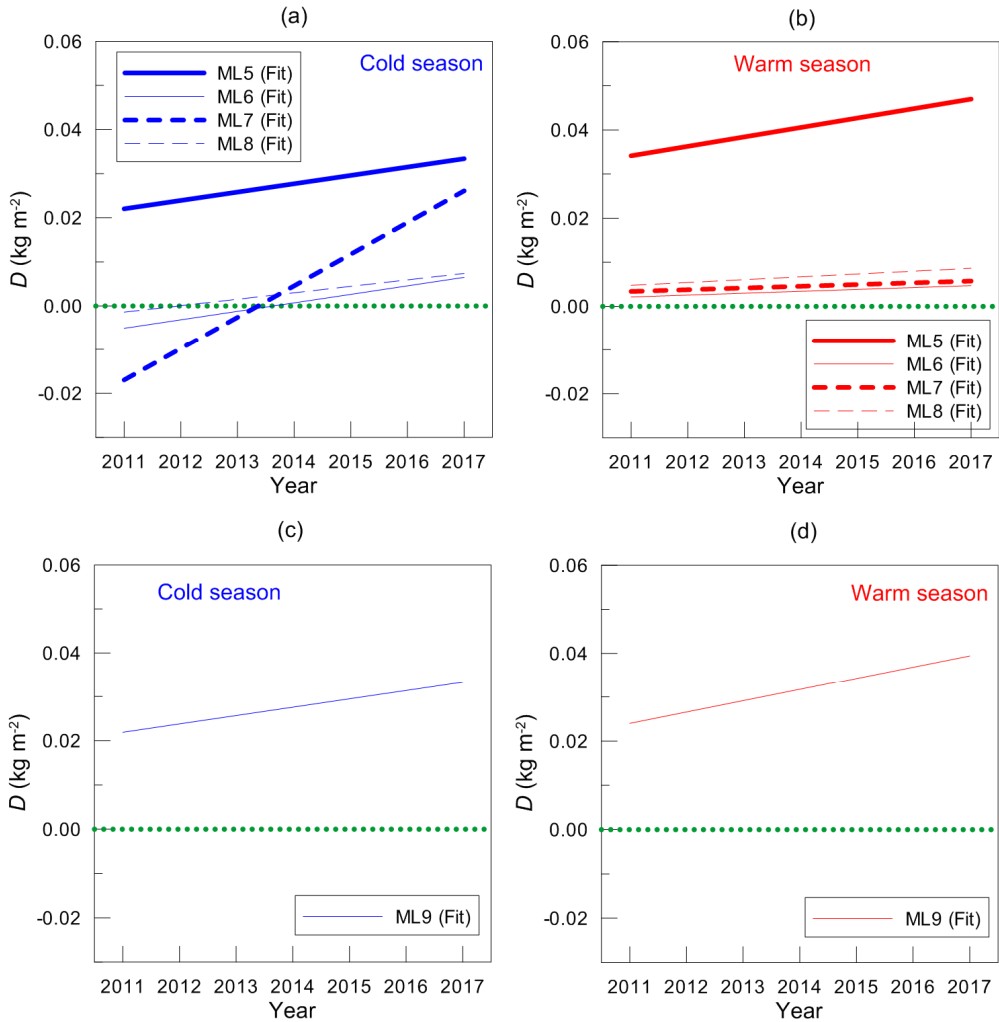

**Figure 9:** Linear fit of the seasonal-mean LWP land-sea gradient within the period 2011-2017 for measurement locations ML5 – ML9 and different seasons. Green dots indicate the zero-gradient line.

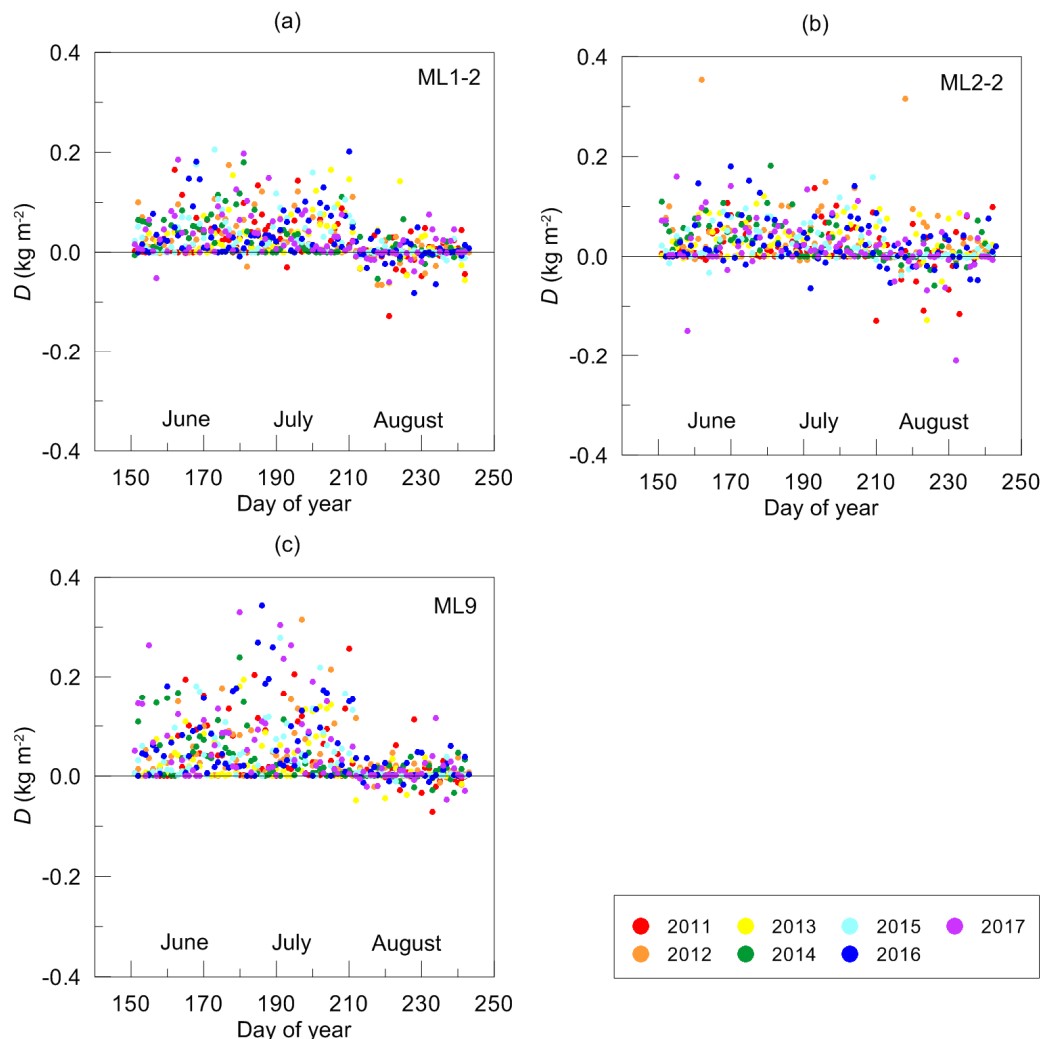

**Figure 10: Intra-seasonal variability of the daily-mean LWP land-sea gradient for measurement locations ML1-2, ML2-2, and ML9 (warm season, seven years of observations – see the legend).**

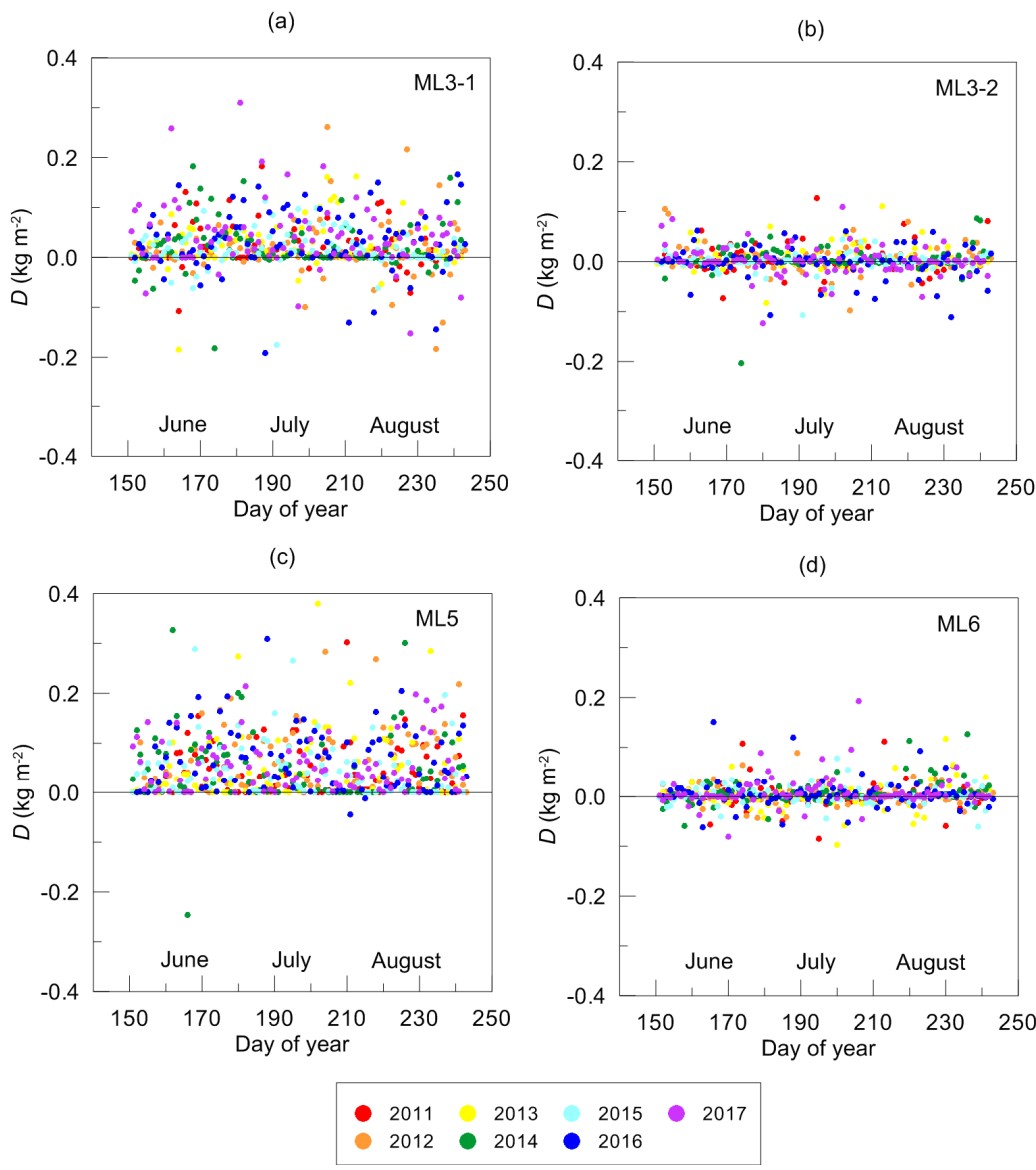

**Figure 11: Intra-seasonal variability of the daily-mean LWP land-sea gradient for measurement locations ML3-1, ML3-2, ML5, and ML9 (warm season, seven years of observations – see the legend).**





**Figure 12: Intra-seasonal variability of the monthly-mean LWP land-sea gradient (warm season, seven years of observations altogether).**





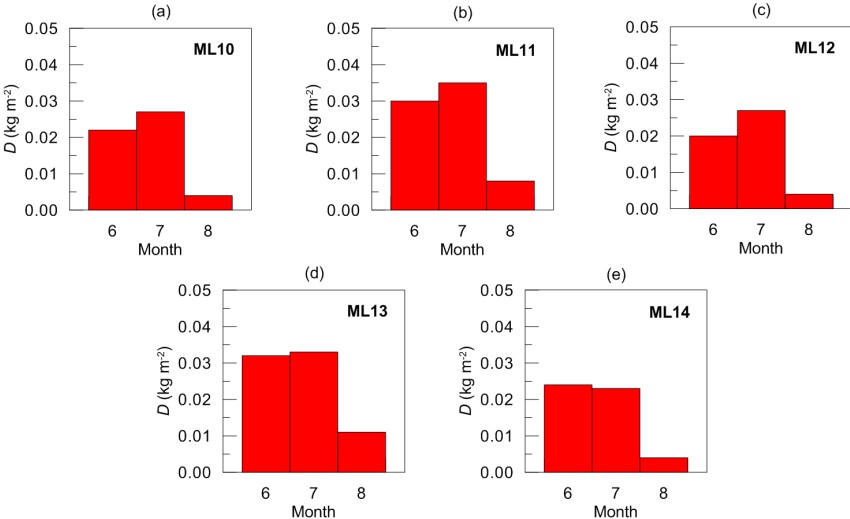

Figure 13: Same as Fig. 12 but for additional measurement locations.





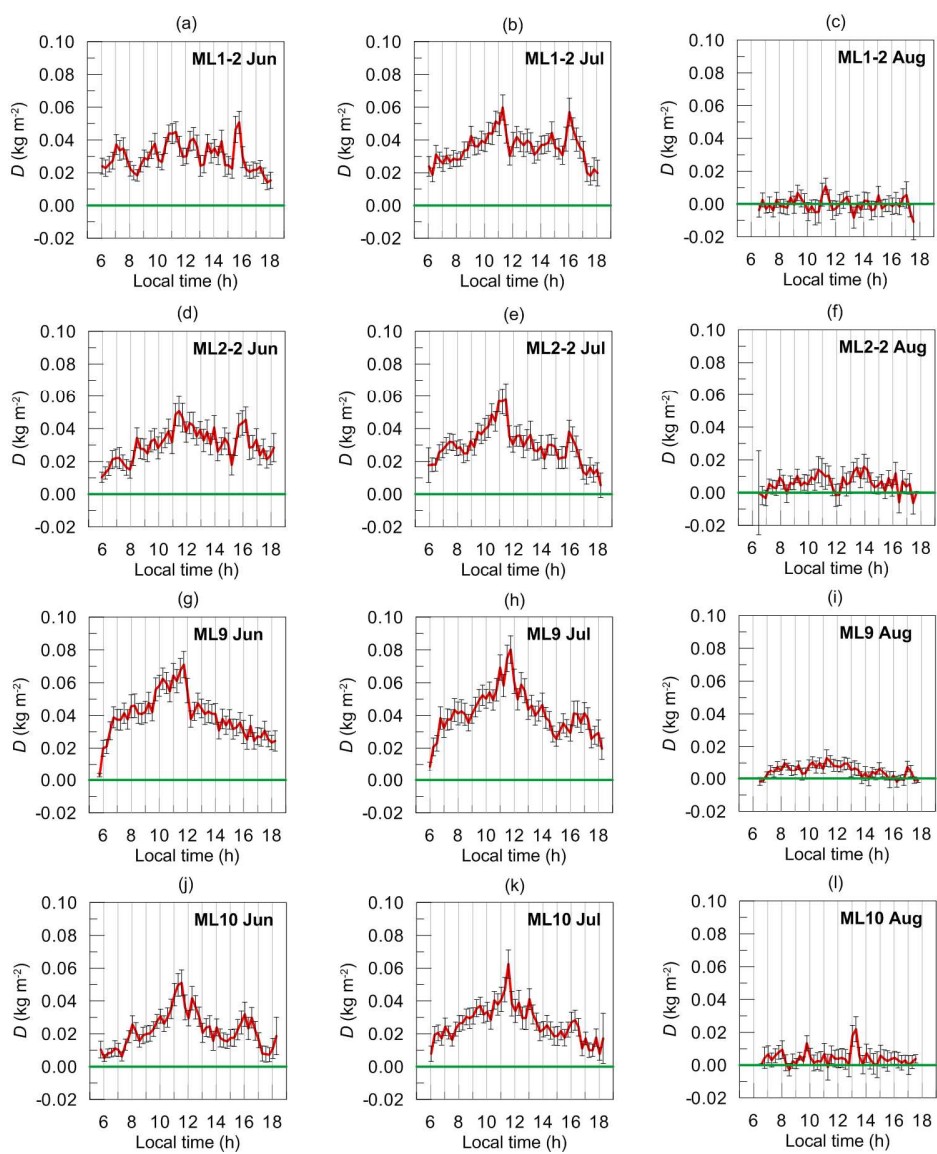

**Figure 14: Diurnal variability of the land-sea LWP gradient at different locations in terms of mean values which refer to time moments of measurements (every 15 minutes). Averaging was done over seven years of observations. Green lines designate zero gradient. Vertical bars demonstrate the standard error of the mean values.**





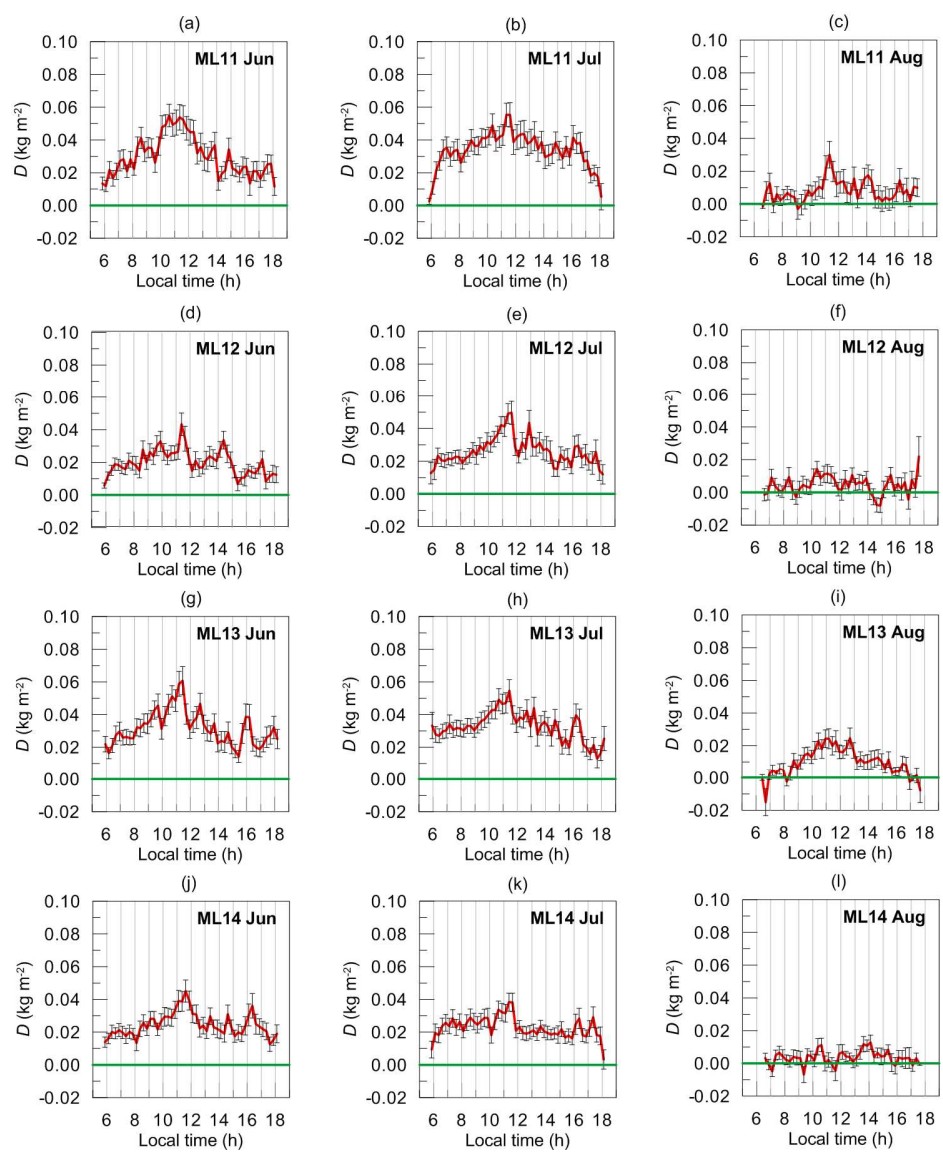

**Figure 15: The same as Fig. 14 but for measurement locations ML11…ML14.**





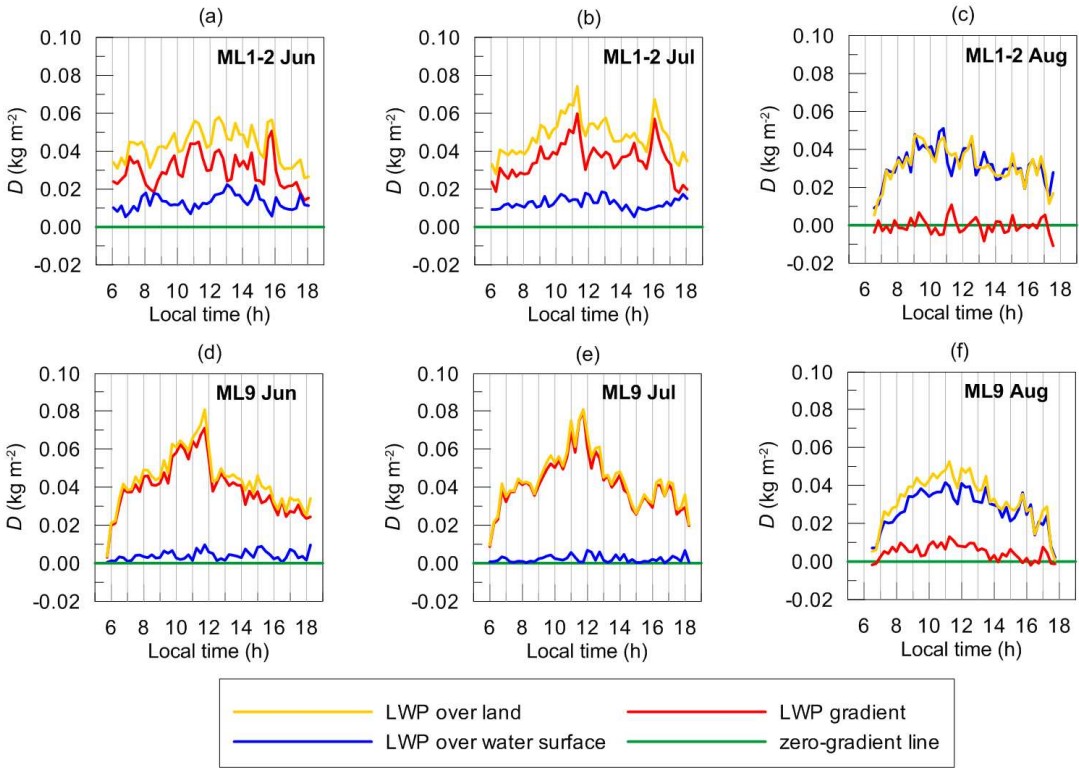

**Figure 16: Diurnal variability of the cloud LWP over land and sea and the land-sea gradient at different locations in terms of mean values which refer to time moments of measurements (every 15 minutes). Averaging was done over seven years of observations.**



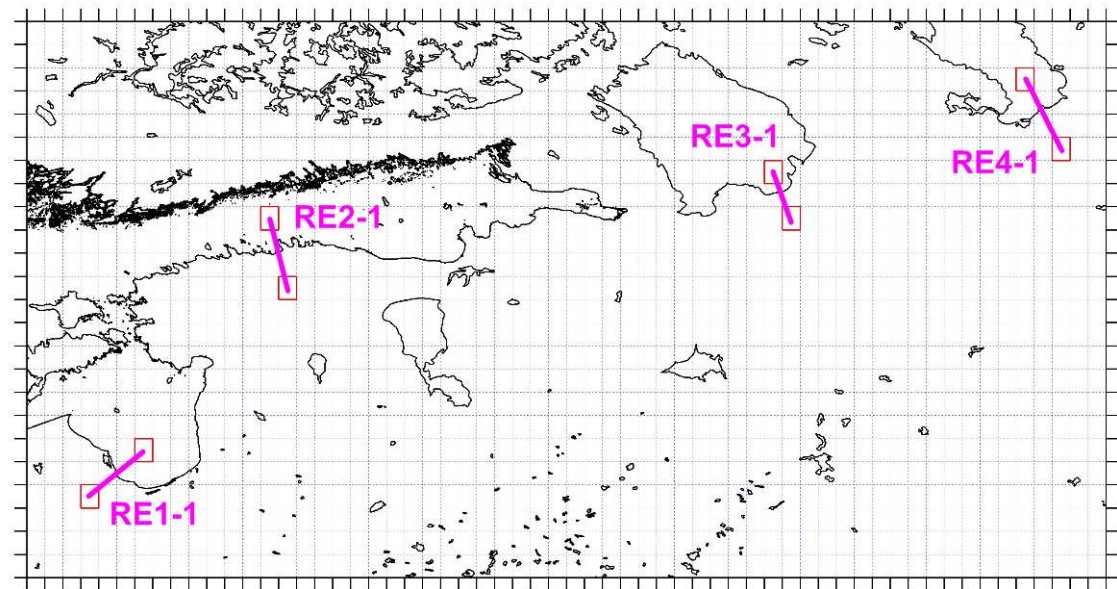

**Figure 17: The map showing the geographical location of the reanalysis grid points used for the calculations of the LWP land-sea gradient. Vector shoreline data: (GSHHG, 2017).**



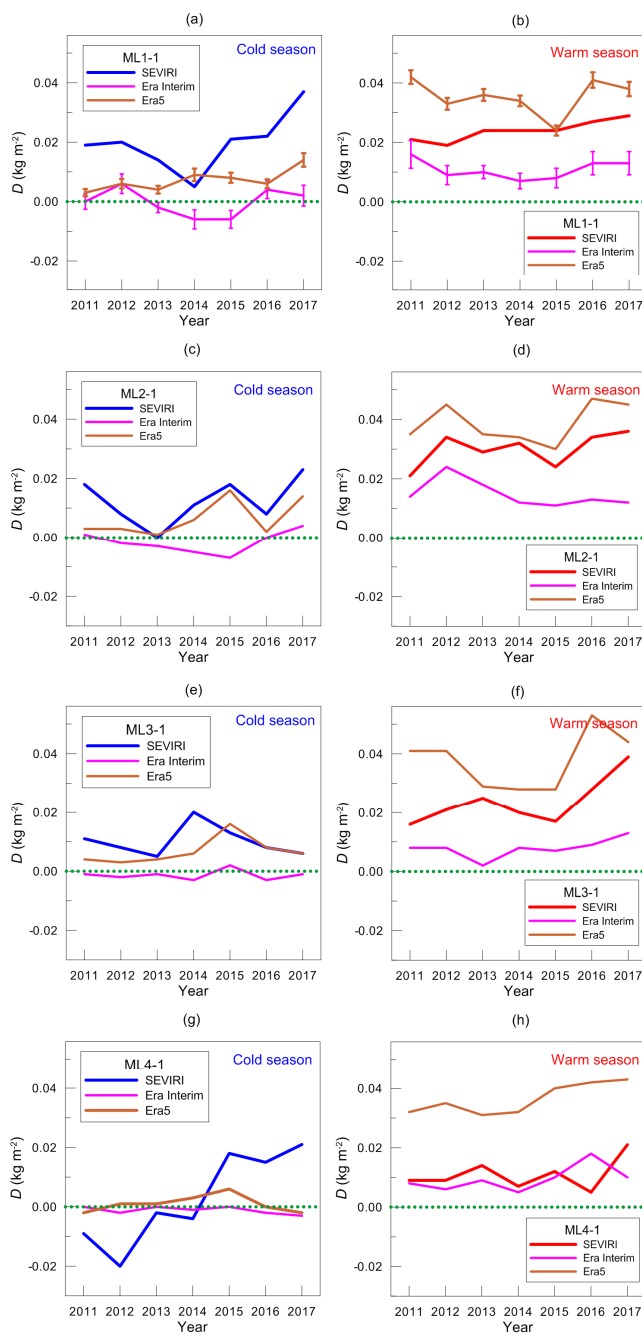

**Figure 18: Inter-annual variability of the seasonal-mean LWP land-sea gradient (large distance) for cold and warm seasons as derived from the SEVIRI measurements and the reanalysis data (Era-Interim and Era5, see the legends). Green dots indicate the zero-gradient line.**





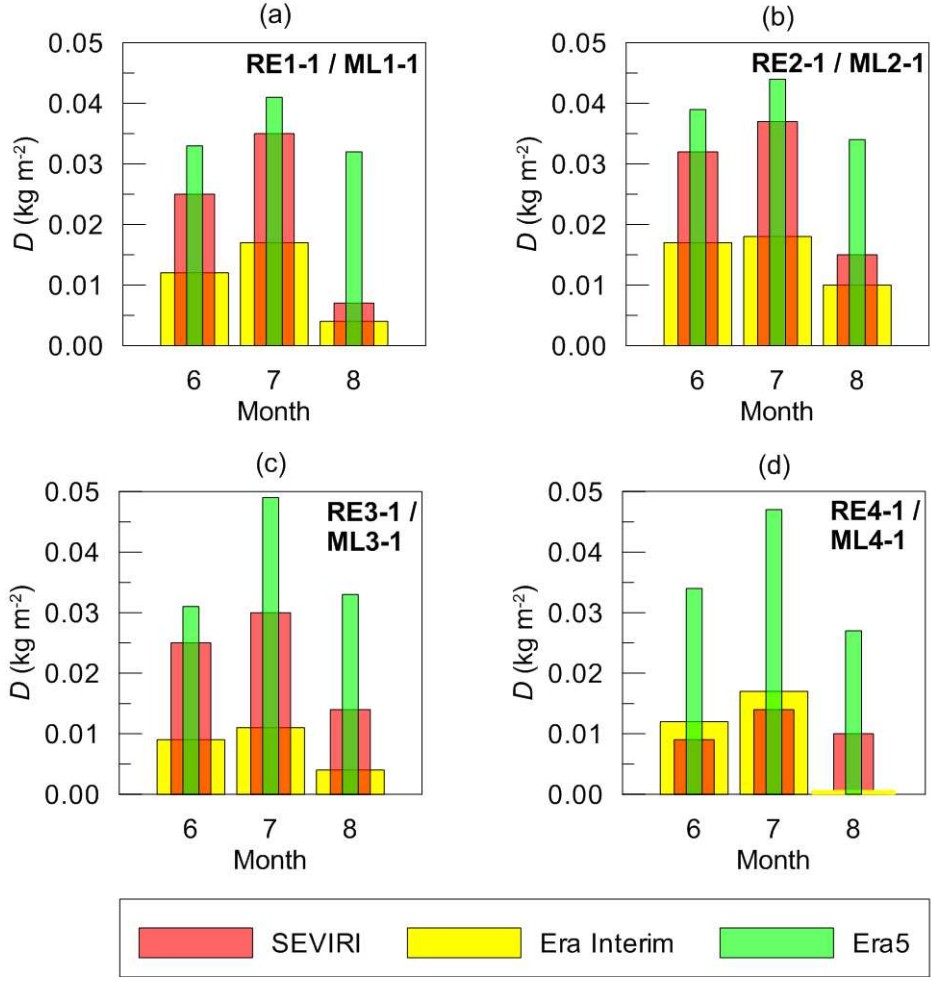

**Figure 19:** Intra-seasonal variability of the monthly-mean LWP land-sea gradient as obtained from the Era-Interim and Era5 reanalyses (warm season, 2011-2017) and the SEVIRI data at the locations which correspond to the large-distance LWP gradients (RE… / ML…).

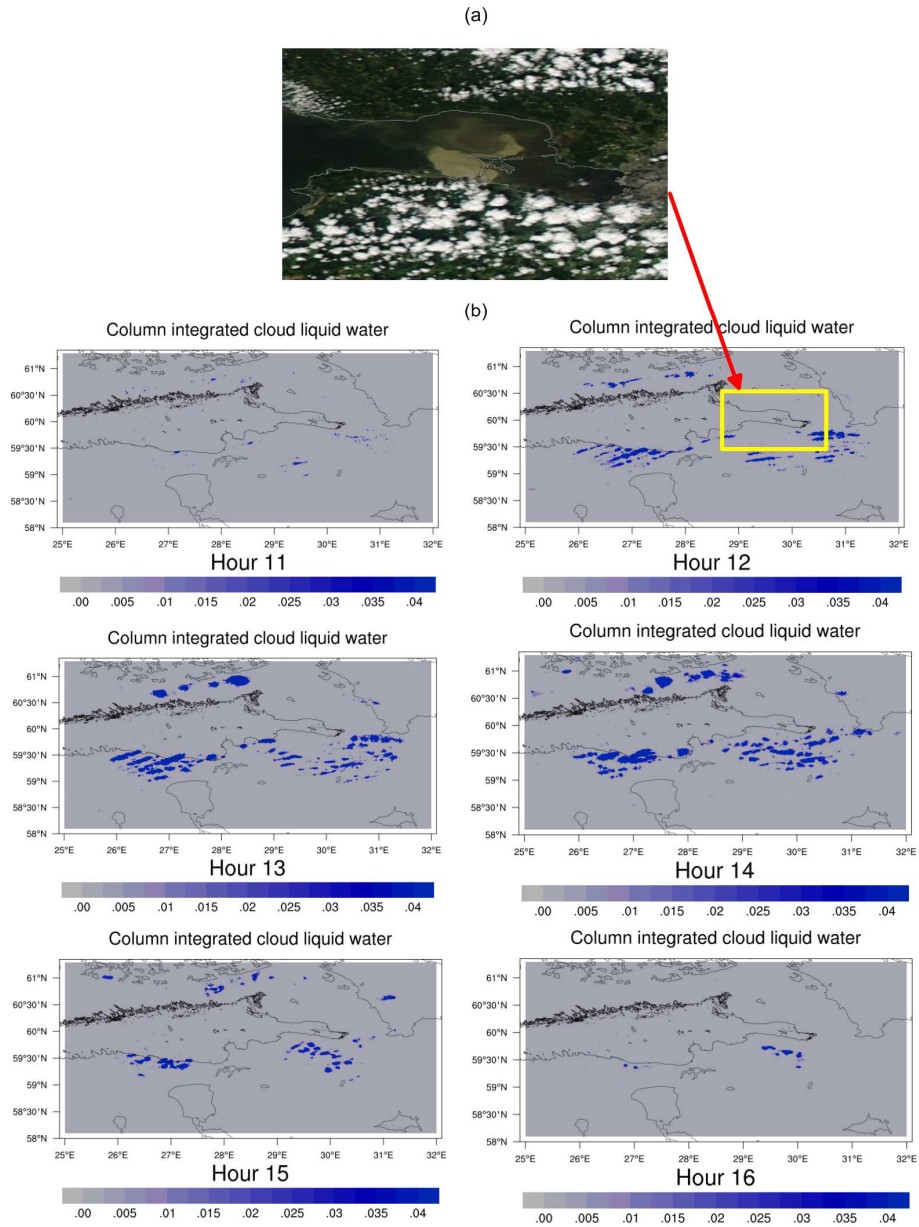

Figure 20. (a) The image of cloud cover acquired by MODIS for 25 July 2015 and for the region inside the considered domain. (b) Cloud liquid water path maps as a result of simulations by the ICON model for 25 July 2015 and for the domain comprising Gulf of Finland and several lakes. Units for LWP (colour bar): kg m$^{-2}$, time: UTC.

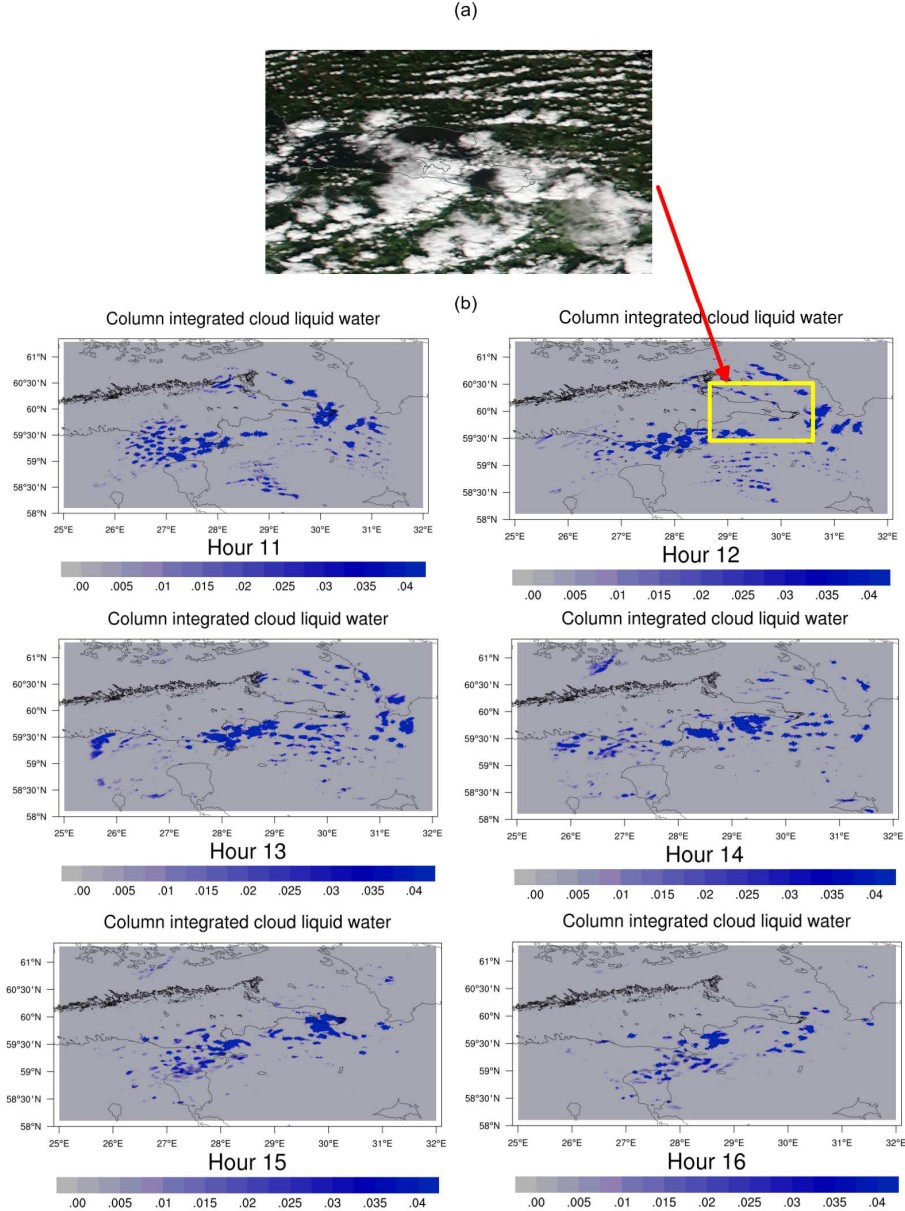

**Figure 21. (a) The image of cloud cover acquired by MODIS for 12 August 2016 and for the region inside the considered domain.**

**(b) Cloud liquid water path maps as a result of simulations by the ICON model for 12 August 2016 and for the domain comprising Gulf of Finland and several lakes. Units for LWP (colour bar): kg m$^{-2}$, time: UTC.**

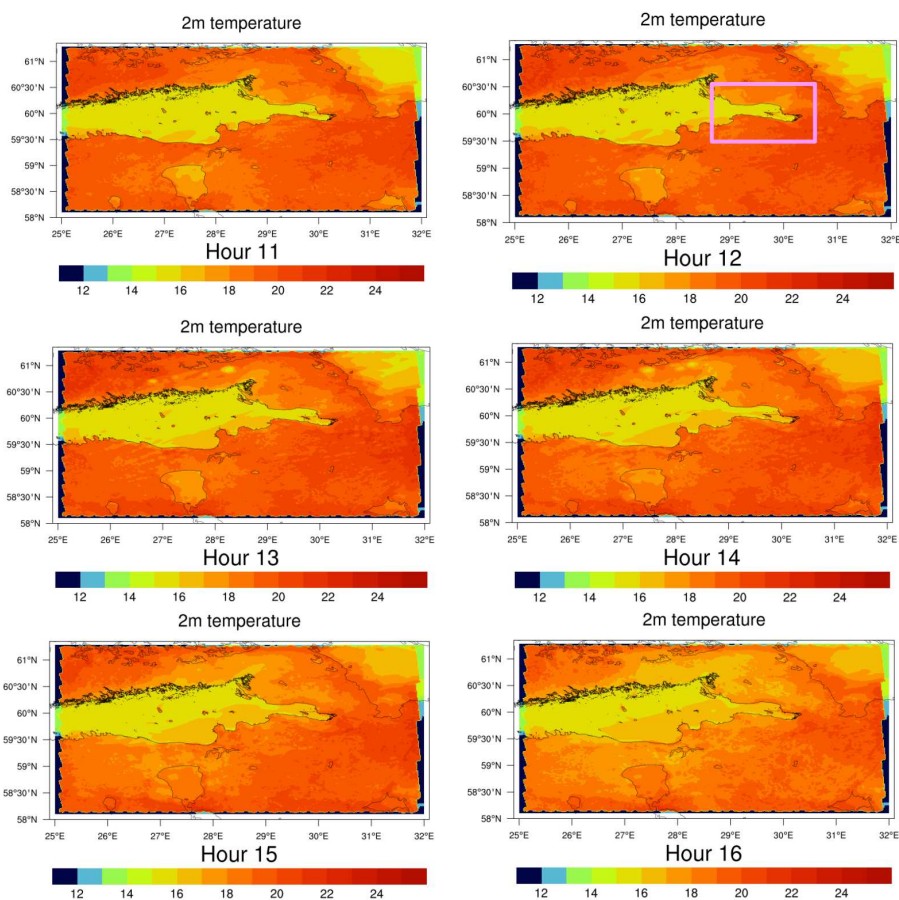

**Figure 22. Surface temperature (at 2 m) maps as a result of simulations by the ICON model for 25 July 2015 and for the domain comprising Gulf of Finland and several lakes. Units for temperature (colour bar): °C, time: UTC. Pink rectangle indicates the territory covered by image from MODIS (see Fig. 20a).**

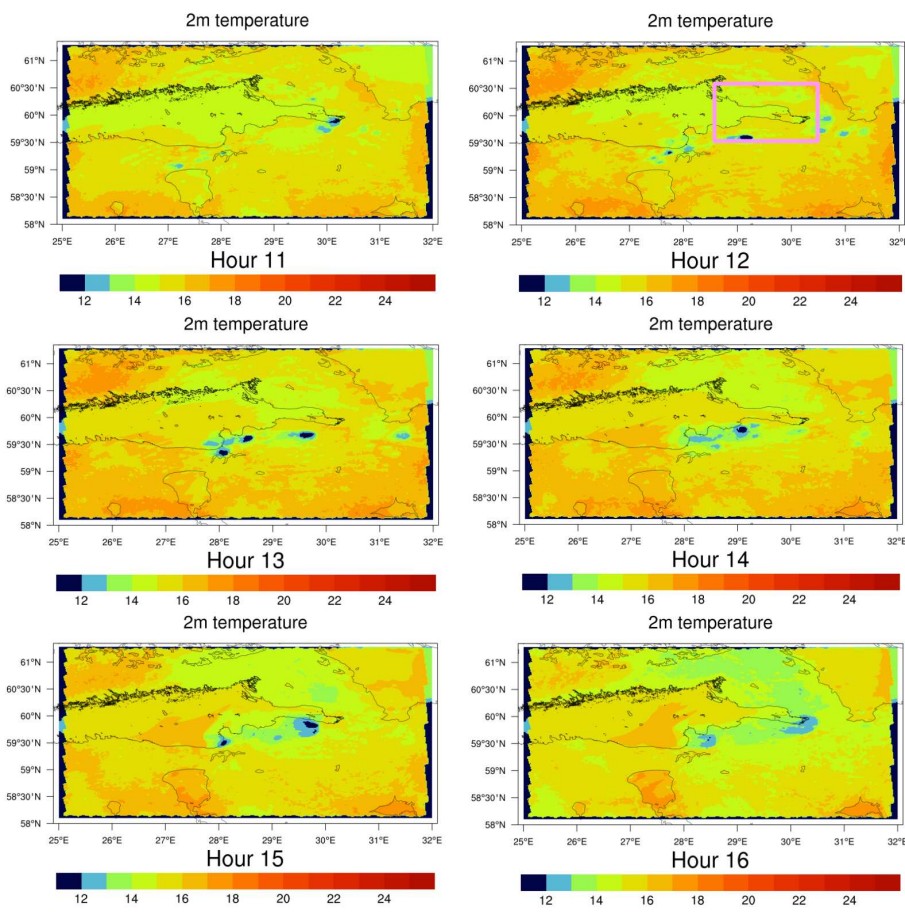

**Figure 23. Surface temperature (at 2 m) maps as a result of simulations by the ICON model for 12 August 2016 and for the domain comprising Gulf of Finland and several lakes. Units for temperature (colour bar): °C, time: UTC. Pink rectangle indicates the territory covered by image from MODIS (see Fig. 20a).**





**Appendix A: Domain-averaged profiles of atmospheric parameters derived from the simulations by ICON**

In order to analyse specific features of the atmospheric state over water and land surfaces, we have selected two sub-domains within the entire domain for modelling and have performed averaging of vertical profiles of several parameters over these sub-domains for both dates: 25 July 2015 and 12 August 2016. One of these sub-domains comprises water surface only and another sub-domain comprises the adjacent land territory, their position is presented in Fig. A1: water pixel 27.0°…27.5° west-east, 59.5°…59.7° south-north; land pixel 27.0°…27.5° west-east, 59.2°…59.4° south-north. We located the sub-domains in such a way that they are undisturbed by smaller land or water bodies. Additionally, the sub-domains contain cloudiness which appeared in both simulated days.. We used the original spatial resolution and plotted the quantities in three-hourly temporal resolution. One should keep in mind that 25 July 2015 has been selected as a typical day with strong LWP land-sea gradient and 12 August has been selected as a typical day without the LWP land-sea gradient. Both days were not affected by a marked frontal passage.

Fig. A2 demonstrates the domain-averaged profiles of cloud cover for both days and Fig. A3 shows specific cloud water content for both days. One can see that on 25 July 2015 there is a weak cloud formation over water, but a considerable cloud amount over land (all around 825 hPa). Maximal values of cloud parameters correspond to 12 h and 15 h UTC both for land and water. In the morning at about 9 h UTC the cloud cover parameter shows cloud formation at the lower layers at about 900 hPa, but this process is not reflected on the water content graph due to negligibly small average values of water content. On 12 August 2016 the cloud cover over land and water resemble each other closely, the diurnal evolution is shifted to later hours over water, as is also the vertical placement of clouds, but the overall evolution is quite comparable. The same behaviour was found for liquid water content. The liquid water content over water is much bigger compared to 25 July 2016.

Fig. A4 demonstrates the domain-averaged profiles of temperature and Fig. A5 shows turbulent kinetic energy for both simulated days. As can be seen from the temperature profiles, during day time from 12 to 15 UTC on 25 July 2015 the atmosphere is rather stably stratified over water, but more unstable over land. The day-night difference in land and water temperature profiles is about 4 K on average, in the near surface layer it is bigger over land by about 4.4 K (3.3 K compared to 7.8). The average profiles of turbulent kinetic energy in Fig. A5 reveal that there is significant turbulence over land during day-time (9…15 h UTC) up to the 825 hPa level, while it is much weaker over water. Mixing and convection take place to a significantly larger extent over land which means moisture can be transported upwards and cloud formation is facilitated. Over water, clouds do not form due to the inhibited convection and may even dissolve around noon when transported to the open water since the relative humidity is low over water. At night-time, turbulence weakens over land, but never ceases completely. Over water, turbulence is stronger in the night hours than during day-time but at all times weaker than over land. In August, the situation changes due to the reduced difference in surface temperature over land and water (compare Figs. 21 and 22). The stratification is not stable over water any longer, but rather unstable or at best neutral as can be seen from the





temperature profiles. There is more turbulence over water during day-time in August, than there was in July. As a result, the atmosphere is well mixed and the relative humidity (not shown) is close to 100 % during day-time in various layers. The profiles over land are unstable from midnight until 18 UTC which facilitates cloud formation and stable afterwards when the sun has set. So, on 12 August 2016 the clouds can sweep from land over to the water area without dissolving or form directly over water.

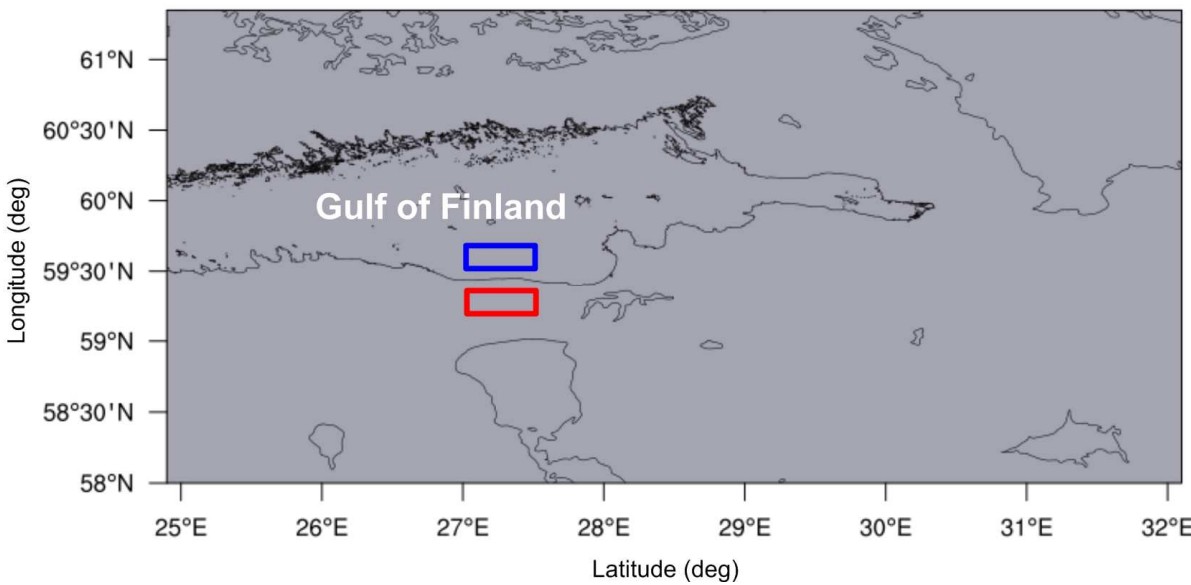

**Figure A1. Two sub-domains for averaging profiles of atmospheric parameters: water surface (blue rectangle) and land surface (red rectangle).**



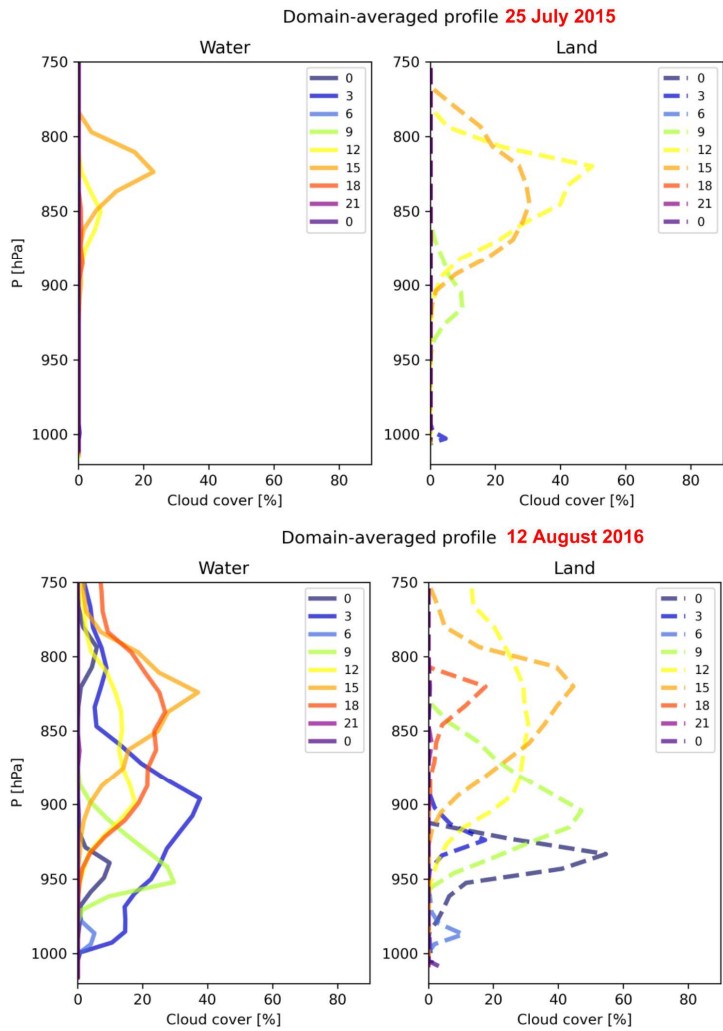

**Figure A2. Domain-averaged profiles of cloud cover for two separately considered domains that comprise all water bodies (left)
and land surface (right): Dates: 25 July 2017 (top) and 12 August 2016 (bottom). Coloured lines correspond to different
time (UTC), see the legend. The entire domain for modelling is shown in Fig. 20.**

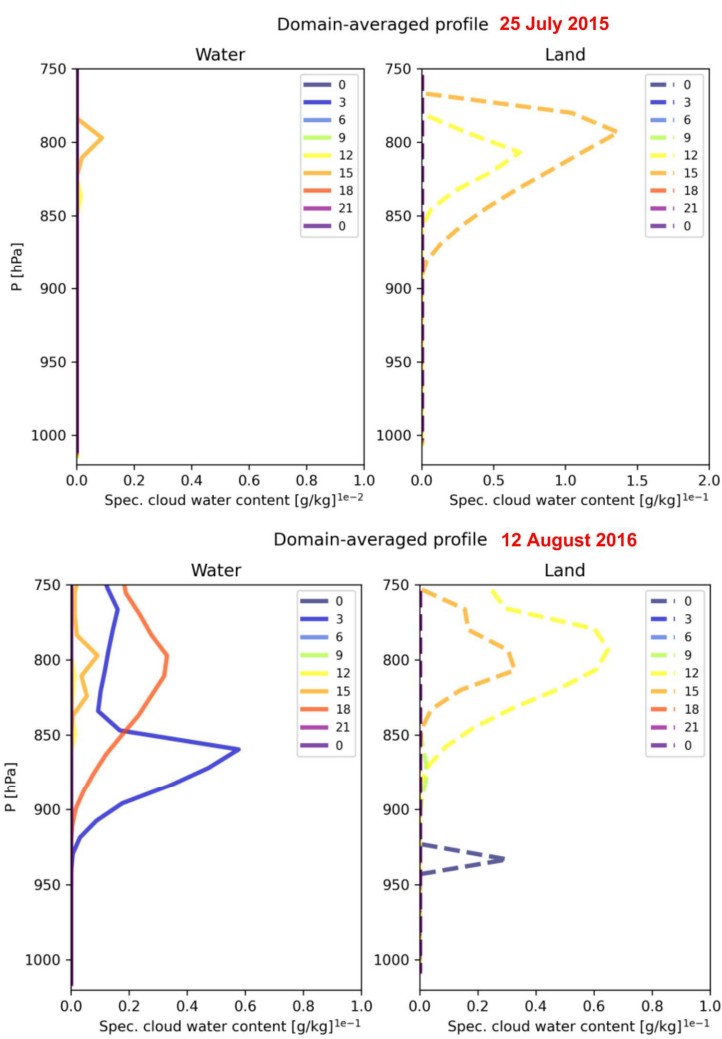

**Figure A3. Same as Fig. A1 but for cloud water content. Please note that horizontal axes have different scale.**

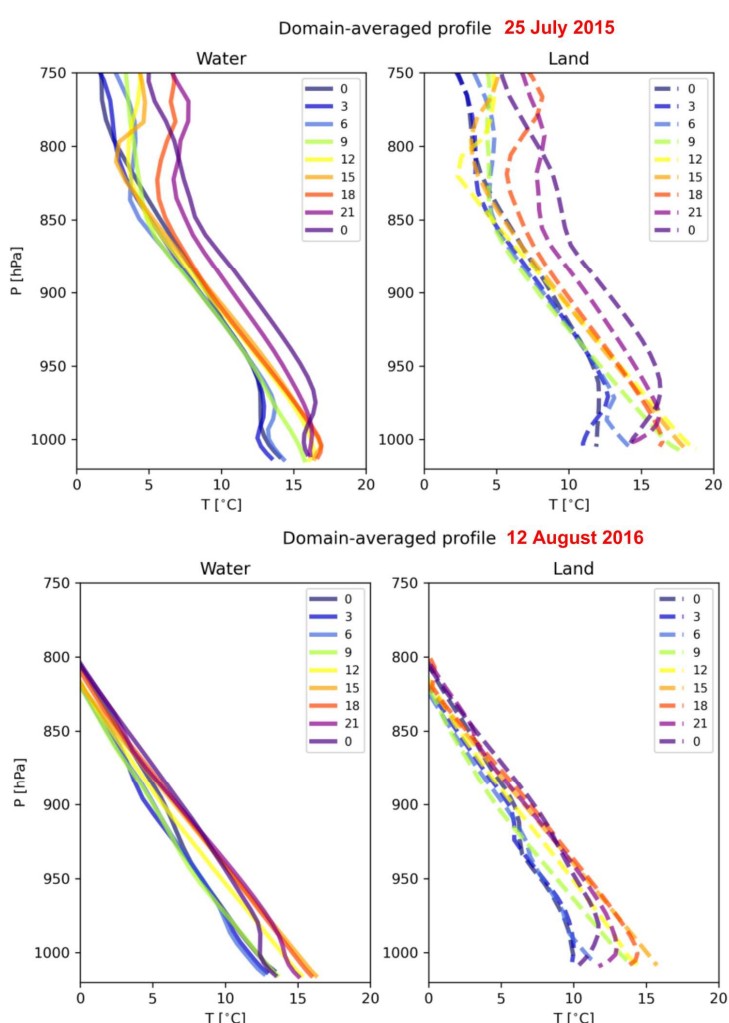

**Figure A4. Same as Fig. A1 but for temperature.**

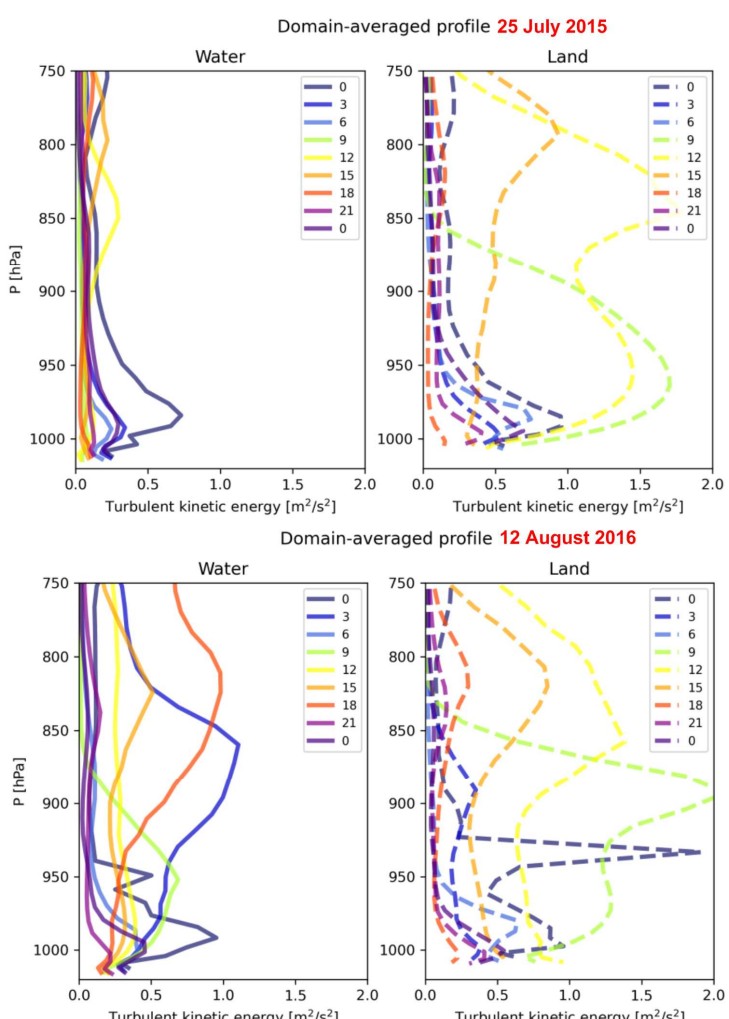

**Figure A5. Same as Fig. A1 but for turbulent kinetic energy.**