# Peer review of "Inter-annual, seasonal and diurnal features of the cloud liquid water path over the land surface and various water bodies in Northern Europe as obtained from the satellite observations by the SEVIRI instrument in 2011-2017"

_Atmospheric Chemistry and Physics, 2021_

## Referee Comment (RC1)

Review for **"Inter-annual, seasonal and diurnal features of the cloud liquid water path over the land surface and various water bodies in Northern Europe as obtained from the satellite observations by the SEVIRI instrument in 2011-2017"** by Kostsov et al., submitted to Atmospheric Chemistry and Physics

**Synthesis:**

This paper presents satellite observations of cloud liquid water path over coastal areas of northern Europe. The authors describe distinct features of cloud cover difference over land and water during different seasons.

**General comments:**

- The motivation of the study and the review of previous work is rather poor. An introduction into typical land-sea contrasts of clouds, wind and temperature and related studies would be necessary in the introduction.

- To me, the word *"gradient"* does not describe what you are analyzing. A gradient is the change of a quantity over a distance. In this case it would be $LWP\ m^{-1}$ > hence the unit would have to be $kg\ m^{-2}\ m^{-1} = kg\ m^{-3}$. This would not be very useful, therefore I suggest to change the wording to *"LWP difference".* Also, your direction of the gradient is wrong. If you call it "*Land-Sea gradient*" your values would have to be negative.

- The limitation to 7 years of data is probably due to large amount of data. However, SEVIRI data are available for the time back to 2004/05. Why didn't you include some years before 2011?

- Trends over 7 years have no statistical significance (e.g. Fig. 9). Excluding just the last year (2017) would already show completely different trends. By using a larger dataset (e.g. 2005-2020) you could test your hypothesis. I can imagine that single outliers can be caused by a dry/wet summer (low/high soil moisture), more or less sea-ice/snow cover, or windy conditions (sea/lake temperatures are less stratified). Therefore, you have to show more proof for your "most important finding" (p.16, l. 474ff and p. 17, l. 515ff) . In the current version, to me there is no proof that another selection of years would not produce totally different trends.

- For the comparison with reanalysis data, I wonder whether the model grid boxes that you chose are really fully placed over sea or land, respectively? In addition, the effective resolution of processes in an atmospheric model is always coarser than the nominal grid spacing.

- I would consider analysing more surface variables for specific days, such as air temperature at some coastal stations, sea surface temperature, diurnal wind patterns (sea breeze), etc. With that, your hypotheses, such as the "August anomaly" could be strongly improved.

- The number of figures should be reduced, or more figures should be combined to one large figure (e.g. Figs. 4-6).

- Please provide scales to your maps (esp. Fig. 17)

Specific comments:

p.3, l. 78: In addition, during winter/spring, (dark) forest areas can absorb considerably more solar radiation than surrounding snow-covered ground or ice-covered water surfaces. This can also lead to updrafts and eventually cloud formation.

p.4, l. 121-122: What do you mean by "simultaneously and not simultaneously"? Do you want to say "cases where both land and water or any of them are clear sky"?

p.10, l. 281 ff: Which time zone did you use for "local time"? UTC+2? If so, please mention it here!

p. 13, l. 380 ff: I am missing some information about the setup of the ICON-LEM model: What is the resolution? Which initial profiles did you use? What is happening at the domain boundaries?

Table 1: What's the percentage of days which were used for the analysis? Is there a significant inter-annual difference?

**Summary**

To summarize, I cannot recommend publication of this manuscript at this stage. The manuscript needs major revisions by carefully considering all the above-mentioned points.

---

## Author Comment (AC1)

**The preliminary reply to the anonymous referee #1 (RC1)**

This is a short preliminary reply to the anonymous referee #1.

We are grateful to the referee for the detailed analysis of our study and for the insightful comments and suggestions. We agree with all statements made by the referee and we will take into account all of them while revising our paper.

Herewith we just would like to make a note on the technical term "gradient" in order to clarify this issue. The referee is perfectly right in his/her statement that we used this term not in its rigorous meaning. Indeed, we investigated two types of the LWP land-sea difference which we called "short distance gradient" and "long distance gradient". This quantities in fact should have had the dimension $kg\,m^2/(20\,km)$ and $kg\,m^2/(80\,km)$. We omitted the denominators for simplicity, in order not to repeat them multiple times. We agree that it was not a good decision.

In the revised version of our paper we will not use the term "gradient". Instead, we will use the terms "difference" and "contrast". The latter one seems to be a good choice. Also, the referee used the term "contrast" himself/herself.

Vladimir Kostsov
(on behalf of all authors)

---

## Author Comment (AC3)

**The reply to the anonymous referee #1 (RC1)**

We are thankful to the referee for the detailed analysis of our study and for the constructive criticism. We agree with most of statements made by the referee and we took into account almost all of them while revising our paper.

Below, the actual comments of the referee are given in **`bold courier font and blue colour`**. The text added to the revised version of the manuscript is marked by red colour.

**`The motivation of the study and the review of previous work is rather poor.`**

We agree with the remark of the esteemed referee that the motivation is presented not enough clear. In the revised version of our manuscript, we restructured the introduction section and organised it as three subsections: Background, Motivation and Novelty. In these subsections we tried to specify all the reasons for doing the study on the LWP land-sea contrast. So, the new section describing motivation is the following:

1.2 Motivation

Primarily, the motivation for our efforts to investigate the LWP land-sea difference originated from our previous studies (Kostsov et al. 2018, 2019) which were devoted to the problem of validation of space-borne remote observations of cloud parameters by means of ground-based passive microwave remote sounding. In these studies microwave measurements were conducted over land but in a coastal area. It should be emphasized that ground-based microwave remote measurements of LWP are the most reliable and widely used tool for validation of observations of LWP from space, in particular by the instruments SEVIRI and AVHRR which measure reflected solar radiation (Roebeling et al., 2008ab; Greuell and Roebeling 2009). However, to the best of our knowledge, there were no validations of space-borne measurements over water areas and over water bodies covered by ice/snow. The importance of such validations arises from the fact that retrieval algorithms use a land-sea mask, and also they use a sea-ice and a snow mask. A misclassification in a mask can cause errors which propagate to higher-level products of the satellite observations. Such situation can occur in winter and during off-season. In winter, the LWP retrieval over highly reflective surfaces (snow and ice) becomes even more complicated problem (Musial et al., 2014), and, as a consequence, the retrieval errors can increase. The mechanism of the error amplification is described by Han et al. (1999) and Platnick et al. (2001): (1) multiple reflections occur between a cloud and underlying surface; (2) the increase in reflectance contributed by a cloud is relatively smaller in case of highly reflective underlying surface. The problem becomes more complicated due to the variability of the ice/snow properties. It has been noted by Platnick et al. (2001) that, as shown in a number of studies, the albedo of the sea ice is dependent on several factors, for example on the presence of air bubbles. Besides, if ice is covered with a snow layer greater than several centimetres the overall reflectance is dominated by this snow layer. Also, the melting process can cause the decreases in reflectance. The complexity of the problem of space-borne remote sensing of cloud parameters over different surfaces stimulated us to conceive the study in which the general features of the LWP land-sea contrast derived from satellite measurements could be summarised and analysed. In our opinion, the joint comprehensive analysis of the large LWP data sets derived from space-borne observations over various surfaces can be valuable for development of validation algorithms.

The importance of studying the LWP land-sea difference rather than the LWP values over land and water separately arises from the fact that inconsistency of data can be detected more easily in this way. The vivid example of detecting inconsistency in data by means of looking at the land-sea contrast of atmospheric parameter is an artefact in ozone column measurements by the TOMS (Total Ozone Mapping Spectrometer) instrument (Cuevas, 2001). Persistent year-to-year differences in total ozone between continents and oceans were found in the mean global ozone data which were averaged in time. This feature has been named GHOST (Global Hidden Ozone Structures from TOMS). Part of these differences appeared to be caused by truncation of the lower tropospheric column due to the topography and by permanent differences in tropopause height distribution. The remaining part (66%) has been found to be an artefact of the retrieval algorithm: the effects of the presence of UV-absorbing

aerosols might have been accounted for not correctly. For examining the effect of each possible contribution to the observed difference, Cuevas (2001) selected the Iberian Peninsula region for a case study. The study by Cuevas (2001) was an encouraging example for us and additional stimulus to investigate common features of the LWP land-sea differences in Northern Europe with the aim to identify the natural effects and possible artefacts in measurements.

The second reason for making the present study was the lack of information on the LWP land-sea differences. Except the above mentioned works by Karlsson there were no special studies focused on the analysis of the LWP values over surfaces of various types in Northern Europe, in particular over land and water areas. Obviously, taking into account the diversity of properties of water bodies and the diversity of the features of local climate, we can expect that the LWP land-sea differences are highly variable in space and time. So far, not enough attention was paid to this interesting issue. In our view, this issue is important for development of regional weather and climate models from the perspective of more accurate simulations over water bodies and in neighbouring areas. As an example, the ICON model can be mentioned which has a special option for weather and climate simulations over lakes (ICON, 2021; ICON Tutorial, 2021).

The third motive to initiate the present study was the fact that so far not much attention was paid to the investigation of physical mechanisms which drive the LWP land-sea differences in Northern Europe. The reason for the differences in spring and summer has been suggested by Karlsson (2003): the inflow of cold water from melting snow and ice is cooling the near-surface atmospheric layer over the water bodies. As a result, in contrast to the land surface, this layer over the water bodies becomes very stable preventing the formation of clouds. This mechanism, however, does not explain the existence of the LWP land-sea difference during cold season when both land and water surfaces are covered with snow and ice. We would like to mention one more mechanism which has been suggested by an expert during an open discussion of the preprint of the present article (https://doi.org/10.5194/acp-2021-387-RC1, last access 29 March 2022):

'In addition, during winter/spring, (dark) forest areas can absorb considerably more solar radiation than surrounding snow-covered ground or ice-covered water surfaces. This can also lead to updrafts and eventually cloud formation.'

The sea breeze mechanism should be mentioned also. Indeed, strong sea breeze fronts initiate vertical currents that are usually marked by the development of cumulus clouds. The detailed review of recent studies of the sea breeze features can be found in the paper by Miller et al. (2003). However the sea breeze mechanism is not able to fully explain the diversity of land-ocean contrasts presented in our work. Indeed, the sea breeze can be the reason for the development of convective cloudiness in the frontal zone, with an inland penetration up to several tens of kilometers. But the results presented in our work demonstrate the systematic suppression of cloudiness over water bodies, with a relatively uniform distribution of cloudiness over the land surface, regardless of the distance from the coastline (see the map in Fig. 2, for example). The sea breeze phenomena certainly can complement another physical mechanism proposed by Karlsson (2003) and already mentioned above. However, both of these mechanisms – the sea breeze circulation and the influx of melt water – cannot explain the existence of the land-ocean contrasts during the cold season, when both land and water surfaces are covered with snow and ice.

In our opinion, the necessary prerequisite for identifying the prevailing physical mechanisms which drive the LWP land-sea differences in Northern Europe is the special detailed statistical analysis of the LWP data provided by the satellite instruments over various water bodies and over land near these water bodies during different seasons. In the present work we make a kind of such analysis.

Added references:

Cuevas, E., Gil, M., Rodriguez, J., Navarro, M., and Hoinka, K.P.: Sea-land total ozone differences from TOMS: GHOST effect, Journal of Geophysical Research, 106 (D21), 27745-27755, https://doi.org/10.1029/2001JD900246, 2001.

Miller, S.T.K., Keim, B.D., Talbot, R.W., Mao, H.: Sea breeze: Structure, forecasting, and impacts, Reviews of Geophysics, 41(3), https://doi.org/10.1029/2003RG000124, 2003.

An introduction into typical land-sea contrasts of clouds, wind and temperature and related studies would be necessary in the introduction.

We do not agree with the statement of the referee that the review of previous work is poor. There is not much information on the LWP land-sea differences in Northern Europe. Except the mentioned works by Karlsson there were no special studies focused on the analysis of the LWP values over surfaces of various types in Northern Europe, in particular over land and water areas. We agree to point that presenting typical land-sea contrasts of cloud parameters and meteorological parameters would be useful for a reader, but such information is absent. Moreover, let's not forget about the variety of water bodies and about the variety of orographic features and local climate of neighbouring land. How can we talk about "typical" values for all diverse cases? And there is one more argument: in fact, to reveal typical features of the LWP land-sea contrast in Northern Europe is exactly the goal of our study.

**To me, the word "gradient" does not describe what you are analyzing. A gradient is the change of a quantity over a distance. In this case it would be LWP m$^{-1}$ > hence the unit would have to be kg m$^{-2}$ m$^{-1}$ = kg m$^{-3}$. This would not be very useful, therefore I suggest to change the wording to "LWP difference". Also, your direction of the gradient is wrong. If you call it "Land-Sea gradient" your values would have to be negative.**

In our preliminary short answer to the referee we have already written that the referee is perfectly right in his/her statement that we used the term "gradient" not in its rigorous meaning. Indeed, we investigated two types of the LWP land-sea difference which we called "short distance gradient" and "long distance gradient". This quantities in fact should have had the dimension kg m$^{-2}$/(20 km) and kg m$^{-2}$/(80 km). We omitted the denominators for simplicity, in order not to repeat them multiple times. We agree that it was not a good decision. In the revised version of our paper we do not use the term "gradient". Instead, we use the terms "difference" and "contrast". The latter one seems to be a good choice since the referee used the term "contrast" himself/herself.

Also, the use of the rigorous term "difference" (or "contrast") helps to avoid ambiguity with the sign of the quantity which is investigated. In the revised version of the paper we define it by Eq.(2) of the manuscript in the following way

$$d = W_{land} - W_{sea}, \qquad (2)$$

where W is the liquid water path value measured in the pixels selected over the land and water areas (as indicated by subscripts).

As a result, the land-sea contrast is expected to be positive, except the abnormal situations.

**The limitation to 7 years of data is probably due to large amount of data. However, SEVIRI data are available for the time back to 2004/05. Why didn't you include some years before 2011?**

Yes, the referee's guess about the large amount of data is correct, it was the first reason. And the second reason for starting with 2011 data was our plans for future research: in a separate study we planned to make comparisons of the LWP land-sea contrast derived from SEVIRI observations with the ground-based data from the HATPRO microwave instrument which started operation only in 2012.

**Trends over 7 years have no statistical significance (e.g. Fig. 9). Excluding just the last year (2017) would already show completely different trends. By using a larger dataset (e.g. 2005-2020) you could test your hypothesis. I can imagine that single outliers can be caused by a dry/wet summer (low/high soil moisture), more or less sea-ice/snow cover, or windy conditions (sea/lake**

**temperatures are less stratified). Therefore, you have to show more proof for your "most important finding" (p.16, l. 474ff and p. 17, l. 515ff) . In the current version, to me there is no proof that another selection of years would not produce totally different trends.**

When we made our study, the most recent version of SEVIRI data was available for 2017 and earlier years. We could not use the data provided by an older version of processing algorithm since the algorithm is constantly improved. At present (March 2022), when preparing the revised version of our paper we have no possibility to extend the data set for analysis due to the case of force majeure: the collaboration between Russian and German scientists is suspended because of current unprecedented tense political situation in the world. It is important to emphasize that our study is the result of collaborative work.

We agree, that our "most important finding" needs more convincing proof. Taking into account the very limited number of data points for multi-year trend analysis, in the revised version we are applying the Fisher criterion for estimating the significance of the linear regression for different locations and seasons. We removed Fig. 9 and added two tables instead, which present the characteristics used for the Fisher criterion. The new part of the Section 3 is the following:

Figs. 7 and 8 provide some indication of a positive temporal trend of the LWP land-sea contrast for several locations. Taking into account the very limited number of data points (only seven) available for multi-year trend analysis, we are applying the Fisher criterion for estimating the statistical significance of the linear regression for different locations and seasons. The algorithm of assessment of statistical significance was taken from the book by Bolshakov (1965). In order to estimate a robustness of a correlation coefficient for a number of data points less than 50 one can use the following function:

$$z = \frac{1}{2}\left[\ln(1+r) - \ln(1-r)\right] \qquad (3)$$

where $r$ is a correlation coefficient. The values of $z$ are normally distributed with the standard deviation:

$$\sigma_z = \frac{1}{\sqrt{n-3}} \qquad (4)$$

where $n$ is the number of data points. In our case $n=7$ and so $\sigma_z=0.5$. For given value of $r$ we calculate the corresponding value of $z$. Then we obtain the values of the correlation coefficient which correspond to the values $z-\sigma_z$ and $z+\sigma_z$ using the inversion of Eq. 3:

$$r_1 = \frac{\exp(2(z-\sigma_z))-1}{\exp(2(z-\sigma_z))+1} \qquad (5a)$$

$$r_2 = \frac{\exp(2(z+\sigma_z))-1}{\exp(2(z+\sigma_z))+1} \qquad (5b)$$

In such a way we obtain the limits of uncertainty of the correlation coefficient:

$$r_1 \leq r \leq r_2 \qquad (6)$$

The linear trend can be considered as statistically significant if the following relation is satisfied:

$$|r| \geq 3\sigma_r \qquad (7)$$

It was shown by Dlin (1958) that the lower limit of the correlation coefficient which satisfies Eq. 7 depends on the number of data points and can be calculated as follows:

$$r_{min} = \frac{\sqrt{n+36}-\sqrt{n}}{6} \qquad (8)$$

In our case $r_{min}=0.65$. Finally, comparing $r_1$ and $r_{min}$ we can find out whether the linear trend is statistically significant:

$$r_1 \geq r_{min} \quad significant$$

$$(9)$$

$$r_1 < r_{min} \quad not \; significant$$

The relations (9) are valid for positive correlation coefficient $r_1$. For negative $r_1$, the minimal absolute value among $r_1$ and $r_2$ should be taken.

Tables 3 and 4 present the results of calculations of correlation coefficients for data pairs "time – LWP contrast" plotted in Figs. 7 and 8 for different locations and seasons. Also, other parameters relevant to assessment of statistical significance of linear trends are given. One can see that for all water bodies except Lake Ladoga the correlation coefficients are positive for both cold and warm seasons. However, there are only four cases which demonstrate robust statistical significance of the linear trend. For cold season, the significant trend is observed for Lake Onega (long-distance LWP difference) and for Lake Ilmen. For warm season, the significant trend is observed for Gulf of Riga (long-distance LWP difference) and for the Neva River Bay. These statistically significant trends are characterised by the following growth rates of the LWP contrast: $0.0064$ kg m$^{-2}$ yr$^{-1}$ (Lake Onega), $0.0072$ kg m$^{-2}$ yr$^{-1}$ (Lake Ilmen), $0.0014$ kg m$^{-2}$ yr$^{-1}$ (Gulf of Riga), $0.0026$ kg m$^{-2}$ yr$^{-1}$ (The Neva River Bay). The rates are larger for cold season. The detected growth rates require confirmation on the basis of expanded datasets. For the time being, no general conclusions could be made.

**Table 3. Parameters ($r$, $r_1$, $r_2$, $r_{min}$) used for assessment of statistical significance (signif.: *yes* or *no*) of linear temporal trend of the LWP land-sea contrast for various locations. Cold season.**

| Water body | Data set | Cold season | | | | |
|---|---|---|---|---|---|---|
| | | $r$ | $r_1$ | $r_2$ | $r_{min}$ | Is trend significant? |
| 1. Gulf of Riga | ML1-1 | 0.52 | 0.08 | 0.79 | | no |
| | ML1-2 | 0.32 | -0.16 | 0.68 | | no |
| 2. Gulf of Finland | ML2-1 | 0.32 | -0.16 | 0.68 | | no |
| | ML2-2 | 0.53 | 0.09 | 0.80 | | no |
| 3. Lake Ladoga | ML3-1 | -0.10 | -0.54 | 0.38 | | no |
| | ML3-2 | 0.56 | 0.13 | 0.81 | | no |
| 4. Lake Onega | **ML4-1** | **0.90** | **0.74** | **0.96** | 0.65 | **yes** |
| | ML4-2 | 0.84 | 0.62 | 0.94 | | no |
| 5. Lake Peipus | ML5 | 0.64 | 0.25 | 0.85 | | no |
| 6. Lake Pihkva | ML6 | 0.43 | -0.04 | 0.75 | | no |
| 7. Lake Ilmen | **ML7** | **0.90** | **0.75** | **0.96** | | **yes** |
| 8. Lake Saimaa | ML8 | 0.31 | -0.18 | 0.68 | | no |
| 9. The Neva River bay | ML9 | 0.33 | -0.16 | 0.69 | | no |

**Table 4. The same as Table 3 but for warm season.**

| Water body | Data set | Warm season | | | | |
|---|---|---|---|---|---|---|
| | | r | $r_1$ | $r_2$ | $r_{min}$ | Is trend significant? |
| 1. Gulf of Riga | **ML1-1** | **0.92** | **0.79** | **0.97** | | **yes** |
| | ML1-2 | 0.69 | 0.33 | 0.87 | | no |
| 2. Gulf of Finland | ML2-1 | 0.55 | 0.12 | 0.81 | | no |
| | ML2-2 | 0.30 | -0.19 | 0.67 | | no |
| 3. Lake Ladoga | ML3-1 | 0.73 | 0.40 | 0.89 | | no |
| | ML3-2 | -0.46 | -0.76 | 0.01 | | no |
| 4. Lake Onega | ML4-1 | 0.38 | -0.10 | 0.71 | 0.65 | no |
| | ML4-2 | 0.02 | -0.45 | 0.48 | | no |
| 5. Lake Peipus | ML5 | 0.53 | 0.08 | 0.79 | | no |
| 6. Lake Pihkva | ML6 | 0.45 | -0.02 | 0.75 | | no |
| 7. Lake Ilmen | ML7 | 0.61 | 0.20 | 0.84 | | no |
| 8. Lake Saimaa | ML8 | 0.65 | 0.27 | 0.86 | | no |
| 9. The Neva River bay | ML9 | **0.92** | **0.79** | **0.97** | | **yes** |

Two references have been added accordingly:

Bolshakov, V.D., Theory of observational errors with basics of probability theory. "Nedra" Publishing, Moscow, 184 P., 1965, (in Russian).

Dlin, A.M., Mathematical statistics in engineering, "Sovetskaya nauka" Publishing, Moscow, 1958, (in Russian).

Taking into account the results of the assessment of the statistical significance of detected trends, we reworded item 3) of the conclusion:

3) The interesting finding is the positive trend of the LWP contrast during 2011-2017 for all considered measurement locations and for both cold and warm seasons with only one exception: Lake Ladoga. Despite the very limited number of data points, the statistical significance of positive linear trends has been confirmed for Lake Onega and Lake Ilmen (cold season) and for Gulf of Riga and the Neva River bay (warm season). These statistically significant trends are characterised by the following growth rates of the LWP contrast: 0.0064 kg m$^{-2}$ yr$^{-1}$ (Lake Onega), 0.0072 kg m$^{-2}$ yr$^{-1}$ (Lake Ilmen), 0.0014 kg m$^{-2}$ yr$^{-1}$ (Gulf of Riga), 0.0026 kg m$^{-2}$ yr$^{-1}$ (The Neva River Bay). The rates are larger for cold season. However, the obtained results require confirmation on the basis of extended data sets before any general conclusions could be made.

We also reworded the last phrases of the paper. The revised text is the following:

Nevertheless, to our opinion, the most interesting findings of the present work are the positive long-term (7-year) trend of the magnitude of the LWP land-sea contrast (statistically significant for four measurement locations) and the so-called "August anomaly": the absence of the LWP difference in August if compared to June and July. It should be emphasised that this "August anomaly" is strictly limited to Gulf of Finland.

Accordingly, we changed the text in the abstract:

The interesting finding is the positive trend of the land-sea LWP difference detected within the time period 2011-2017 which appeared to be statistically significant for four water bodies (lakes Onega and Ilmen, Gulf of Riga and the Neva River bay).

**For the comparison with reanalysis data, I wonder whether the model grid boxes that you chose are really fully placed over sea or land, respectively? In addition, the effective resolution of processes in an atmospheric model is always coarser than the nominal grid spacing.**

Spatial resolution of the reanalysis data is already discussed in detail in Section 6, second paragraph:
*"In the present study we consider the ERA-Interim and Era5 reanalyses..."*
Position of grid points is shown in Fig. 17. In order to avoid a situation when a reader can be misled, we explicitly indicate in caption and in the text that grid points Fig. 17 are relevant to Era5. We also make a note in the text that grid boxes of Era5 have been selected in a way to be fully placed over sea or land:

It should be noted that grid boxes of Era5 have been selected in a way to be fully placed over sea or land. It was possible since long distance LWP differences are considered.

And we mention that for Era Interim, due to coarser original spatial resolution, the original grid boxes (before interpolation to 28km-grid) can contain a small portion of the "wrong" surface: sea for a land grid box and land for a sea grid box:

One should keep in mind that for Era Interim, due to coarser original spatial resolution, the original grid boxes (before interpolation to 28km-grid) can contain a small portion of the "wrong" surface: sea for a land grid box and land for a sea grid box.

Also, we placed a scale (100 km bar) in Fig. 17 and added the information to axes:

[Figure]

Figure 17: The map showing the geographical location of the Era5 reanalysis grid points used for the calculations of the LWP land-sea contrast. Vector shoreline data: (GSHHG, 2017).

I would consider analysing more surface variables for specific days, such as air temperature at some coastal stations, sea surface temperature, diurnal wind patterns (sea breeze), etc. With that, your hypotheses, such as the "August anomaly" could be strongly improved.

We are thankful to the referee for this insightful comment. Indeed, the "August anomaly" was quite unexpected phenomenon which required explanation. When we made the present study and revealed this feature we made some attempts to investigate it in more detail. However, it appeared that these activities required separate quite large amount of efforts, in particular the efforts to analyse surface variables, as the esteemed referee suggests. Such work seemed to go beyond the scope of the present study. Therefore, we decided to limit our efforts by several model runs to have the impression how the LWP land-sea contrasts are reproduced. But there was also another reason to postpone the analysis of surface variables. After completing the present study, we intensively worked on the problem of assessment of the LWP land-sea contrast by ground-based spectral-angular microwave observations at the coastline of the Neva River bay. There was a hope that this work could provide a direct proof of this feature and give a new big stimulus for detailed investigation of the observed effects, in particular "August anomaly" which was observed by the SEVIRI instrument at this location also. Now this work is completed and the preprint has been published in Atmospheric Measurement Techniques Discussions:

*Kostsov, V., Ionov, D., and Kniffka, A.: Retrieval of the land-sea contrast of cloud liquid water path by applying a physical inversion algorithm to combined zenith and off-zenith ground-based microwave measurements, Atmos. Meas. Tech. Discuss. [preprint], https://doi.org/10.5194/amt-2021-415, in review, 2022.*

The effect of "August anomaly" was not confirmed by the ground-based observations. Therefore, at present we are coming round to the opinion that "August anomaly" can be an artefact of the SEVIRI measurements. The argument in favour of this hypothesis is, first, the immediacy of the transition from large contrast to zero contrast and, second, the fact that this transition occurs exactly in the beginning of August at all examined locations in the Gulf of Finland. In the end of Section 4 we added some speculations about this possible reason of the anomaly:

> The revealed effect can be called "August anomaly". The similarity of results obtained for different locations in the Baltic Sea can lead to the conclusion about possible common physical mechanisms that drive the LWP land-sea difference in the entire Baltic Sea region considered in the present study. In order to explain this effect one can suggest collecting and analysing sets of surface variables, such as air temperature at some coastal stations, sea surface temperature, diurnal wind patterns (sea breeze), etc. However, such data search and analysis seem to form separate quite large research activity which goes beyond the scope of the present study. Therefore, in the present study we limit our efforts by several model runs to have the impression how the LWP land-sea contrasts are reproduced (see Section 7 below).

> One more important notice should be made. While the preprint of the present article was in review during an open discussion phase after submission to the journal "Atmospheric Chemistry and Physics", we intensively worked on the problem of assessment of the LWP land-sea contrast by ground-based spectral-angular microwave observations at the coastline of the Neva River bay. In this separate work, the physical inversion algorithm was used for processing ground-based measurements. Earlier, we used the regression algorithm (Kostsov et al., 2020). Subsequently, the physical inversion algorithm was selected as more accurate, if compared to regression approach, and the most suitable for error estimation and quality control. There was a hope that this work and new more accurate results could provide a direct independent proof of "August anomaly" which was clearly observed by the SEVIRI instrument at the location of ground-based microwave measurements (the Neva River bay). Now this work is completed and the preprint has been published in Atmospheric Measurement Techniques Discussions (Kostsov et al., 2022). The effect of "August anomaly" was not confirmed by the ground-based observations. Therefore, at present we

are coming round to the opinion that "August anomaly" can be an artefact of the SEVIRI measurements. The arguments in favour of this hypothesis are, first, the immediacy of the transition from large LWP contrast to very low contrast (just within a couple of days) and, second, the fact that this transition occurs exactly in the beginning of August at all examined locations in the Gulf of Finland. It is unlikely that natural meteorological processes change so sharply and synchronically at different places in the Baltic Sea. So, we assume that "August anomaly" can be an artefact which reflects certain algorithmic features in the SEVIRI data. If this hypothesis were confirmed, then we would have the situation similar to situation with the TOMS instrument described in the Introduction section when the observed features of the land-sea difference in total ozone turned out to be an artefact and helped to identify an error and to correct a data processing algorithm.

**The number of figures should be reduced, or more figures should be combined to one large figure (e.g. Figs. 4-6).**

Following the advice of the referee we combined Figs. 4-6 to one figure which shows only typical distributions of the LWP land-sea contrast:

[Figure]

Figure 4: Typical statistical distributions (in terms of relative frequency of occurrence R) of the LWP land-sea contrast values for different measurement locations and seasons. Please note that for better visibility the vertical axes are broken and have different scaling in the lower and upper part.

The text which describes the distributions has been changed accordingly.

Fig. 9 has been removed. Instead, the results of the assessment of statistical significance of trends have been added (already described above).

Figs. 10 and 11 have been combined in one Fig. 7, only typical cases are demonstrated now:

[Figure]

Figure 7: Intra-seasonal variability of the daily-mean LWP land-sea contrast for measurement locations ML1-2, ML2-2, and ML9 (warm season, seven years of observations – see the legend).

The text has been changed accordingly.

We removed Fig. 15 where diurnal features of the LWP contrast are similar to features presented in Fig. 14 and mentioned in the text that the only one exception from common behaviour is the location ML13 in August.

Finally, we managed to decrease the number of Figures by 5: from 23 to 18.

**Please provide scales to your maps (esp. Fig. 17)**

In the revised version the scales (100 km bars) are provided in all figures with maps with only few exceptions for the maps generated by the ICON model on the latitude-longitude grid with equal steps in degrees. In these maps, distance between objects may be slightly distorted which does not influence the quality of demonstration of the results.

**Specific comments:**

**p.3, l. 78: In addition, during winter/spring, (dark) forest areas can absorb considerably more solar radiation than surrounding snow-covered ground or ice-covered water surfaces. This can also lead to updrafts and eventually cloud formation.**

We are thankful to the referee for this remark and we included it in the text:

> Also, we would like to mention one more mechanism which has been suggested by an expert during an open discussion of the preprint of the present article (https://doi.org/10.5194/acp-2021-387-RC1, last access 29 March 2022):
>
> > *'In addition, during winter/spring, (dark) forest areas can absorb considerably more solar radiation than surrounding snow-covered ground or ice-covered water surfaces. This can also lead to updrafts and eventually cloud formation.'*

**p.4, l. 121-122: What do you mean by "simultaneously and not simultaneously"? Do you want to say "cases where both land and water or any of them are clear sky"?**

Yes, exactly. We corrected the text in the following way:

> The data selected for analysis include all clear sky cases where both land and water or any of them are clear sky.

**p.10, l. 281 ff: Which time zone did you use for "local time"? UTC+2? If so, please mention it here!**

We are grateful to the referee for this remark. Indeed, we missed to clarify the important issue about time that we used. We added the following information in the beginning of Section 5 (diurnal features):

> It is important to note that for each location the time of SEVIRI measurements was converted to local solar time using geographical longitude. It means that for each location the 12 h moment on the time scale exactly corresponds to local noon.

**p. 13, l. 380 ff: I am missing some information about the setup of the ICON-LEM model: What is the resolution? Which initial profiles did you use? What is happening at the domain boundaries?**

We added this information to Section 7:

> For each single day, ICON was running with a global setting and a refined nest over the study region with the horizontal resolution of about 2.5 km and triangular grid. The study region is shown in Fig. 15b. A border zone of about 30 km of the study region was excluded from the analysis since it was used as a nudging zone for the lateral boundary data. These data are needed to force the large scale flow on the limited aread grid once per hour and are stored as global fields. The reliable input data for modelling were taken from archives which were available for 2015 and later. Modelling of a single day required processing of about 130 Gbyte of input data. The high resolution limited area mode which we used for simulations is not suitable for climate time scale studies, therefore we simulated single days only.

In order just to give the impression about the resolution, in the plot below we present the temperature of the lowest model level on the original triangular grid:

[Figure]

Since the article is already nearly overloaded with plots, we did not add this figure in the revised version.

**Table 1: What's the percentage of days which were used for the analysis? Is there a significant inter-annual difference?**

We are grateful to the referee for the advice to check the percentage of days used for the analysis. Indeed, it should be done in order to identify possible data gaps which may influence the results of the analysis. We calculated for each month the percentage of days which were used for the analysis. The results are presented below in two Tables for warm and cold seasons:

**Table Percentage1. The percentage of days which were used for the analysis. Calculations for each month. Warm season. Data sets (locations) ML1-ML9.**

| Data set / Month | ML1-1 | ML1-2 | ML2-1 | ML2-2 | ML3-1 | ML3-2 | ML4-1 | ML4-2 | ML5 | ML6 | ML7 | ML8 | ML9 |
|---|---|---|---|---|---|---|---|---|---|---|---|---|---|
| Jun-2011 | 97 | 100 | 100 | 100 | 93 | 93 | 87 | 90 | 100 | 100 | 97 | 97 | 97 |
| Jul-2011 | 87 | 90 | 97 | 97 | 90 | 97 | 100 | 100 | 100 | 100 | 100 | 100 | 100 |
| Aug-2011 | 94 | 97 | 87 | 100 | 97 | 94 | 97 | 97 | 97 | 100 | 94 | 90 | 94 |
| Jun-2012 | 97 | 100 | 93 | 97 | 90 | 97 | 93 | 93 | 93 | 100 | 93 | 93 | 97 |
| Jul-2012 | 97 | 97 | 97 | 100 | 97 | 100 | 100 | 100 | 97 | 94 | 97 | 94 | 94 |
| Aug-2012 | 90 | 97 | 87 | 94 | 97 | 97 | 90 | 94 | 100 | 100 | 97 | 90 | 97 |
| Jun-2013 | 97 | 100 | 100 | 100 | 97 | 100 | 97 | 97 | 97 | 100 | 100 | 93 | 97 |
| Jul-2013 | 94 | 100 | 100 | 100 | 94 | 100 | 100 | 94 | 100 | 100 | 100 | 97 | 100 |
| Aug-2013 | 94 | 100 | 90 | 97 | 90 | 94 | 97 | 97 | 100 | 100 | 100 | 97 | 100 |
| Jun-2014 | 100 | 100 | 93 | 93 | 97 | 97 | 90 | 97 | 97 | 93 | 100 | 83 | 90 |
| Jul-2014 | 94 | 94 | 100 | 100 | 97 | 97 | 94 | 100 | 97 | 100 | 97 | 97 | 100 |
| Aug-2014 | 87 | 90 | 81 | 90 | 94 | 94 | 97 | 90 | 84 | 100 | 90 | 94 | 97 |
| Jun-2015 | 93 | 100 | 97 | 100 | 100 | 100 | 100 | 100 | 100 | 100 | 100 | 97 | 100 |
| Jul-2015 | 97 | 94 | 97 | 97 | 97 | 94 | 94 | 97 | 94 | 97 | 97 | 94 | 97 |
| Aug-2015 | 100 | 100 | 100 | 100 | 97 | 97 | 97 | 97 | 97 | 100 | 97 | 97 | 100 |
| Jun-2016 | 90 | 97 | 87 | 97 | 93 | 97 | 100 | 100 | 93 | 100 | 100 | 100 | 93 |
| Jul-2016 | 97 | 100 | 94 | 100 | 87 | 90 | 90 | 94 | 94 | 97 | 94 | 90 | 100 |
| Aug-2016 | 94 | 100 | 90 | 97 | 97 | 97 | 97 | 97 | 94 | 97 | 94 | 84 | 87 |
| Jun-2017 | 93 | 97 | 90 | 93 | 90 | 93 | 87 | 87 | 90 | 100 | 90 | 83 | 87 |
| Jul-2017 | 97 | 97 | 94 | 94 | 100 | 100 | 90 | 97 | 90 | 97 | 94 | 94 | 94 |
| Aug-2017 | 97 | 97 | 87 | 90 | 90 | 97 | 94 | 100 | 97 | 90 | 100 | 90 | 94 |

**Table Percentage2. The percentage of days which were used for the analysis. Calculations for each month. Cold season. Data sets (locations) ML1-ML9.**

| Data set \ Month | ML1-1 | ML1-2 | ML2-1 | ML2-2 | ML3-1 | ML3-2 | ML4-1 | ML4-2 | ML5 | ML6 | ML7 | ML8 | ML9 |
|---|---|---|---|---|---|---|---|---|---|---|---|---|---|
| Feb-2011 | 64 | 71 | 43 | 46 | 36 | 39 | 25 | 25 | 50 | 61 | 57 | 29 | 36 |
| Mar-2011 | 90 | 94 | 87 | 90 | 90 | 94 | 84 | 71 | 94 | 94 | 94 | 97 | 97 |
| Feb-2012 | 55 | 55 | 41 | 38 | 34 | 38 | 24 | 24 | 38 | 55 | 55 | 28 | 34 |
| Mar-2012 | 94 | 97 | 90 | 90 | 90 | 94 | 90 | 94 | 94 | 94 | 97 | 97 | 94 |
| Feb-2013 | 64 | 64 | 36 | 39 | 21 | 32 | 14 | 14 | 39 | 61 | 57 | 18 | 29 |
| Mar-2013 | 94 | 94 | 97 | 100 | 94 | 90 | 90 | 90 | 97 | 90 | 94 | 97 | 100 |
| Feb-2014 | 39 | 43 | 29 | 32 | 21 | 32 | 11 | 11 | 36 | 54 | 50 | 18 | 29 |
| Mar-2014 | 84 | 90 | 71 | 81 | 71 | 77 | 68 | 71 | 77 | 90 | 87 | 81 | 77 |
| Feb-2015 | 61 | 64 | 32 | 32 | 29 | 29 | 21 | 21 | 46 | 64 | 54 | 25 | 43 |
| Mar-2015 | 84 | 87 | 81 | 87 | 74 | 81 | 81 | 81 | 87 | 90 | 90 | 94 | 84 |
| Feb-2016 | 55 | 59 | 28 | 31 | 31 | 38 | 24 | 28 | 38 | 55 | 55 | 24 | 34 |
| Mar-2016 | 87 | 87 | 90 | 94 | 90 | 87 | 81 | 81 | 87 | 94 | 87 | 77 | 94 |
| Feb-2017 | 50 | 50 | 21 | 25 | 21 | 21 | 18 | 25 | 36 | 46 | 46 | 21 | 29 |
| Mar-2017 | 81 | 84 | 84 | 90 | 90 | 97 | 84 | 87 | 87 | 84 | 77 | 97 | 87 |

One can see that for June, July and August (warm season) and for March (cold season) and all years of observations there are no data gaps and there is no any noticeable inter-annual difference for all locations. For February (cold season), the percentage considerably differs from location to location, but for a single location the inter-annual difference is not considerable. In order not to overload the article, we decided not to include these tables in the revised version. Instead, we added the following text in the revised version of our article after the analysis of Table 1 in the end of Section 2:

In order to identify possible data gaps which may influence the results of the analysis we calculated for each month the percentage of days which were used for the analysis. These calculations have shown that for June, July and August (warm season) and for March (cold season) and all years of observations there are no data gaps and there is no any noticeable inter-annual difference for all locations. Also, there is no any noticeable difference between the calculated quantities for different locations: they are mostly within the interval 80%-100%. For February (cold season), the percentage considerably differs from location to location within the interval 10-70%, but for a single location the inter-annual difference is not considerable.

In conclusion, we would like to thank once again the referee for valuable remarks and suggestions which we sincerely appreciate.

**Vladimir Kostsov,**
corresponding author

---

## Author Comment (AC4)

**The reply to the anonymous referee #2 (RC2)**

We are grateful to the referee for the remarks. We took them into account while revising our paper. However, we do not agree with several suggestions made by the esteemed referee and below we present our argumentation for that.

Below, the actual comments of the referee are given in **`bold courier font and blue colour`**. The text added to the revised version of the manuscript is marked by red colour.

**`This manuscript focuses on the variability of the LWP gradient by seasons and by the horizontal scale of lakes.`**

To our opinion, the esteemed referee's view on the problem considerably differs from our view. We selected the first statement of the referee as the key which can help to understand and explain the reason for that. The statement presented above is correct, but we should not forget about the origin of experimental data taken for analysis. Actually, the origin of data and the reliability of data are very important issues which should be kept in mind before making any analysis.

We confess that may be it was our fault not to make proper emphasis on the fact that the LWP data derived from the space-borne observations by the SEVIRI instrument over water are still not validated and therefore may contain errors. Therefore, the analysis of the variability of the LWP land-sea contrast had the aim not only to reveal specific seasonal and diurnal features, but also to identify possible artefacts in measurements. It is obvious, that assessment of the self consistency of data should be of higher priority with respect to analysis of possible physical reasons of detected seasonal and diurnal features of the quantity under consideration (the LWP land-sea contrast in our case).

We guess that the referee assumes that our goal is to find explanations for the observed variability of the LWP contrast. It is true, but this goal stands in the relatively long perspective. The task of higher priority is to identify the main features of the LWP land-sea contrast and to eliminate possible measurement errors and artefacts. Fulfilling this task has appeared to be a separate large study which we presented in our manuscript. Therefore, in order to clarify our view on the problem and explicitly declare the goals and novelty of our study, we added the subsections "1.2 Motivation" and "1.3 Novelty" in the Introduction.

**`Obviously, the variability depends on the dynamic and thermodynamic states around the lakes (oceans). It is impossible to avoid the analysis of the boundary layer structure and comparison of the characteristic length-scale of the circulation and the scale of lakes. Linear theory will support you to explain the observed phenomena. However, the authors did not mention the dynamical aspect of the meso-scale circulation in the introduction section at all and slightly looked at the ICON simulations.`**

We have carefully studied the referee's comment containing a strong proposal to include in the analysis of the observed phenomena the "dynamical aspect of the meso-scale circulation". We agree that the introduction section and discussion of the identified land-ocean contrasts lacks a mention of the sea breeze mechanism. Indeed, strong sea breeze fronts initiate vertical currents that are often marked by the development of cumulus clouds. In response to this referee's remark, in the revised version we mention the sea breeze mechanism in the Subsection 1.2. We are grateful to the referee for providing links to relevant studies, which we reviewed in detail. However, our attention was drawn to another, later work by Miller et al. (2003) which provides a comprehensive review of see breeze research dating back 2500 years and focuses on recent studies. Therefore, we decided to refer specifically to this article in the revised version of our

paper rather than to the papers suggested by the esteemed referee. And we emphasize, that modeling and applying the sea breeze mechanism to our results is definitely beyond the scope of our present study.

Explanation of the motivation for our study and mentioning the sea breeze mechanism have been included in the revised version as follows:

**1.2 Motivation**

Primarily, the motivation for our efforts to investigate the LWP land-sea difference originated from our previous studies (Kostsov et al. 2018, 2019) which were devoted to the problem of validation of space-borne remote observations of cloud parameters by means of ground-based passive microwave remote sounding. In these studies microwave measurements were conducted over land but in a coastal area. It should be emphasized that ground-based microwave remote measurements of LWP are the most reliable and widely used tool for validation of observations of LWP from space, in particular by the instruments SEVIRI and AVHRR which measure reflected solar radiation (Roebeling et al., 2008ab; Greuell and Roebeling 2009). However, to the best of our knowledge, there were no validations of space-borne measurements over water areas and over water bodies covered by ice/snow. The importance of such validations arises from the fact that retrieval algorithms use a land-sea mask, and also they use a sea-ice and a snow mask. A misclassification in a mask can cause errors which propagate to higher-level products of the satellite observations. Such situation can occur in winter and during off-season. In winter, the LWP retrieval over highly reflective surfaces (snow and ice) becomes even more complicated problem (Musial et al., 2014), and, as a consequence, the retrieval errors can increase. The mechanism of the error amplification is described by Han et al. (1999) and Platnick et al. (2001): (1) multiple reflections occur between a cloud and underlying surface; (2) the increase in reflectance contributed by a cloud is relatively smaller in case of highly reflective underlying surface. The problem becomes more complicated due to the variability of the ice/snow properties. It has been noted by Platnick et al. (2001) that, as shown in a number of studies, the albedo of the sea ice is dependent on several factors, for example on the presence of air bubbles. Besides, if ice is covered with a snow layer greater than several centimetres the overall reflectance is dominated by this snow layer. Also, the melting process can cause the decreases in reflectance. The complexity of the problem of space-borne remote sensing of cloud parameters over different surfaces stimulated us to conceive the study in which the general features of the LWP land-sea contrast derived from satellite measurements could be summarised and analysed. In our opinion, the joint comprehensive analysis of the large LWP data sets derived from space-borne observations over various surfaces can be valuable for development of validation algorithms.

The importance of studying the LWP land-sea difference rather than the LWP values over land and water separately arises from the fact that inconsistency of data can be detected more easily in this way. The vivid example of detecting inconsistency in data by means of looking at the land-sea contrast of atmospheric parameter is an artefact in ozone column measurements by the TOMS (Total Ozone Mapping Spectrometer) instrument (Cuevas, 2001). Persistent year-to-year differences in total ozone between continents and oceans were found in the mean global ozone data which were averaged in time. This feature has been named GHOST (Global Hidden Ozone Structures from TOMS). Part of these differences appeared to be caused by truncation of the lower tropospheric column due to the topography and by permanent differences in tropopause height distribution. The remaining part (66%) has been found to be an artefact of the retrieval algorithm: the effects of the presence of UV-absorbing aerosols might have been accounted for not correctly. For examining the effect of each possible contribution to the observed difference, Cuevas (2001) selected the Iberian Peninsula region for a case study. The study by Cuevas (2001) was an encouraging example for us and additional stimulus to investigate common features of the LWP land-sea differences in Northern Europe with the aim to identify the natural effects and possible artefacts in measurements.

The second reason for making the present study was the lack of information on the LWP land-sea differences. Except the above mentioned works by Karlsson there were no special studies focused on the analysis of the LWP values over surfaces of various types in Northern Europe, in particular over land and water areas. Obviously, taking into account the diversity of properties of water bodies and the diversity of the features of local climate, we can expect that the LWP land-sea differences are highly variable in space and time. So far, not enough attention was paid to this interesting issue. In our view, this issue is important for development of regional weather and climate models from the perspective of

more accurate simulations over water bodies and in neighbouring areas. As an example, the ICON model can be mentioned which has a special option for weather and climate simulations over lakes (ICON, 2021; ICON Tutorial, 2021).

The third motive to initiate the present study was the fact that so far not much attention was paid to the investigation of physical mechanisms which drive the LWP land-sea differences in Northern Europe. The reason for the differences in spring and summer has been suggested by Karlsson (2003): the inflow of cold water from melting snow and ice is cooling the near-surface atmospheric layer over the water bodies. As a result, in contrast to the land surface, this layer over the water bodies becomes very stable preventing the formation of clouds. This mechanism, however, does not explain the existence of the LWP land-sea difference during cold season when both land and water surfaces are covered with snow and ice. We would like to mention one more mechanism which has been suggested by an expert during an open discussion of the preprint of the present article (https://doi.org/10.5194/acp-2021-387-RC1, last access 29 March 2022):

'In addition, during winter/spring, (dark) forest areas can absorb considerably more solar radiation than surrounding snow-covered ground or ice-covered water surfaces. This can also lead to updrafts and eventually cloud formation.'

The sea breeze mechanism should be mentioned also. Indeed, strong sea breeze fronts initiate vertical currents that are usually marked by the development of cumulus clouds. The detailed review of recent studies of the sea breeze features can be found in the paper by Miller et al. (2003). However the sea breeze mechanism is not able to fully explain the diversity of land-ocean contrasts presented in our work. Indeed, the sea breeze can be the reason for the development of convective cloudiness in the frontal zone, with an inland penetration up to several tens of kilometers. But the results presented in our work demonstrate the systematic suppression of cloudiness over water bodies, with a relatively uniform distribution of cloudiness over the land surface, regardless of the distance from the coastline (see the map in Fig. 2, for example). The sea breeze phenomena certainly can complement another physical mechanism proposed by Karlsson (2003) and already mentioned above. However, both of these mechanisms – the sea breeze circulation and the influx of melt water – cannot explain the existence of the land-ocean contrasts during the cold season, when both land and water surfaces are covered with snow and ice.

In our opinion, the necessary prerequisite for identifying the prevailing physical mechanisms which drive the LWP land-sea differences in Northern Europe is the special detailed statistical analysis of the LWP data provided by the satellite instruments over various water bodies and over land near these water bodies during different seasons. In the present work we make a kind of such analysis.

Added references:

Cuevas, E., Gil, M., Rodriguez, J., Navarro, M., and Hoinka, K.P.: Sea-land total ozone differences from TOMS: GHOST effect, Journal of Geophysical Research, 106 (D21), 27745-27755, https://doi.org/10.1029/2001JD900246, 2001.

Miller, S.T.K., Keim, B.D., Talbot, R.W., Mao, H.: Sea breeze: Structure, forecasting, and impacts, Reviews of Geophysics, 41(3), https://doi.org/10.1029/2003RG000124, 2003.

**Since the corresponding author already documented several papers about the land-ocean contrast, and hence, it is about time to analyze dynamics in addition to the statistical analyses. Therefore, my recommendation is reject.**

We would like to make some clarification. So far, we published three papers:

Kostsov, V. S., Kniffka, A., and Ionov, D. V.: Cloud liquid water path in the sub-Arctic region of Europe as derived from ground-based and space-borne remote observations, Atmos. Meas. Tech., 11, 5439-5460, doi:10.5194/amt-11-5439-2018, 2018.

Kostsov, V. S., Kniffka, A., Stengel, M., and Ionov, D. V.: Cross-comparison of cloud liquid water path derived from observations by two space-borne and one ground-based instrument in

northern Europe, Atmos. Meas. Tech., 12, 5927–5946, https://doi.org/10.5194/amt-12-5927-2019, 2019.

Kostsov, V. S., Ionov, D. V., and Kniffka, A.: Detection of the cloud liquid water path horizontal inhomogeneity in a coastline area by means of ground-based microwave observations: feasibility study, Atmos. Meas. Tech., 13, 4565–4587, https://doi.org/10.5194/amt-13-4565-2020, 2020.

In the first two papers the problem of the assessment of the LWP land-sea contrast was just shortly mentioned. These papers were sharply focused on the comparison of the space-borne and ground-based measurements of LWP. So, up to now we made only one study completely focused on the LWP land-sea contrast (Kostsov et al., 2020) but this study was devoted to the problem of the LWP contrast detection only from ground-based observations at only one location. One can see that our present work is just the second specialized study of the LWP land-sea contrast and the very first study of its spatial and temporal features. Therefore, we do not agree with the referee's statement that the time has come to perform the analysis of dynamic processes. We also would like to mention that actually we started such analysis but using the ICON model. In our opinion, it is the matter of authors choice what tool and what way to use for investigations. We decided to follow the way of using modern state-of-the-art weather and climate models which are able to simulate a bunch of different processes and at the same time to account for local orography. The above mentioned progress of the present work in comparison to our previous studies is summarized in the revised version in Section 1.3:

**1.3 Novelty**

In the present study, the focus is made on the temporal and spatial variations of LWP in coastal areas at different scales. The goal of the present study is to analyse the phenomenon of the LWP horizontal inhomogeneities in the vicinity of a number of water bodies in Northern Europe which differ significantly in their geomorphology (shape, area, volume, etc.): Gulf of Finland, Gulf of Riga, the Neva River bay, Lake Ladoga, Lake Onega, Lake Peipus, Lake Pihkva, Lake Ilmen, and Lake Saimaa. The study is based on LWP data over Northern Europe obtained from seven years (2011-2017) of the space-borne measurements by the SEVIRI instrument. Initially, our aim was to answer the following main questions:

- What are the statistical distributions of the LWP land-sea difference during different seasons at different water bodies?
- Does the LWP land-sea contrast always exist during warm and cold season at water bodies with different properties, and what is its magnitude for large and small water bodies?
- How strong is the inter-annual variability of the LWP land-sea contrast and are there any long-term trends?
- Are there any characteristic features in the diurnal variations of the LWP land-sea contrast (for the day time when space-borne measurements by SEVIRI are available)?
- Is there any correlation between the ice/snow cover period and the magnitude of the LWP contrast?
- Can we distinguish artefacts in the LWP contrast data provided by SEVIRI and, if yes, when and how often do these artefacts appear?

In addition, for several specific cases, atmospheric parameters over the mesoscale domain comprising Gulf of Finland and several lakes have been simulated with the numerical model ICON in limited area and weather prediction mode. The goal of these simulations was to assess how modern state-of-the-art weather-climate model which account for a variety of processes and produce self-consistent data can be used for studying the problem of formation of the LWP land-sea contrast.

The authors can easily find some past researches on the meso-scale circulation related to the land-ocean contrast as follows. Please review in detail.
Hadi et al., (2000) Tropical Sea-breeze Circulation and Related Atmospheric Phenomena Observed with L-band Boundary Layer Radar in Indonesia, https://www.jstage.jst.go.jp/article/jmsj1965/78/2/78_2_123/_article

Niino (1987)The Linear Theory of Land and Sea Breeze Circulation,
https://www.jstage.jst.go.jp/article/jmsj1965/65/6/65_6_901/_article/-char/en

Yan and Anthes (1987) The Effect of Latitude on the Sea Breeze
https://doi.org/10.1175/1520-0493(1987)115<0936:TEOLOT>2.0.CO;2

We are grateful to the referee for providing these references. We have carefully read these papers which will certainly help us in future specialised research after the LWP contrast data will be validated and possible artefacts will be removed. To our opinion, making focus on sea breezes can be misleading at the present step of investigations.

**Vladimir Kostsov,**
corresponding author